# Induction of apoptosis in human colorectal cancer cells by nanovesicles from fingerroot (*Boesenbergia rotunda* (L.) Mansf.)

Saharut Wongkaewkhiaw[1], Amaraporn Wongrakpanich[2], Sucheewin Krobthong[3], Witchuda Saengsawang[1,3,4], Arthit Chairoungdua[1,4,5,6], Nittaya Boonmuen[1]*

1 Department of Physiology, Faculty of Science, Mahidol University, Bangkok, Thailand, 2 Department of Pharmacy, Faculty of Pharmacy, Mahidol University, Bangkok, Thailand, 3 Center for Neuroscience, Faculty of Science, Mahidol University, Bangkok, Thailand, 4 Excellent Center for Drug Discovery (ECDD), Mahidol University, Bangkok, Thailand, 5 Toxicology Graduate Program, Faculty of Science, Mahidol University, Bangkok, Thailand, 6 Center of Excellence on Environmental Health and Toxicology, OPS, MHESI, Bangkok, Thailand

* nittaya.bom@mahidol.ac.th

**Data Availability Statement:** All relevant data are within the paper and its Supporting information files.

## Abstract

Colorectal cancer is the leading cause of cancer-related deaths worldwide, warranting the urgent need for a new treatment option. Plant-derived nanovesicles containing bioactive compounds represent new therapeutic avenues due to their unique characteristics as natural nanocarriers for bioactive molecules with therapeutic effects. Recent evidence has revealed potential anticancer activity of bioactive compounds from *Boesenbergia rotunda* (L.) Mansf. (fingerroot). However, the effect and the underlying mechanisms of fingerroot-derived nanovesicles (FDNVs) against colorectal cancer are still unknown. We isolated the nanovesicles from fingerroot and demonstrated their anticancer activity against two colorectal cancer cell lines, HT-29 and HCT116. The IC$_{50}$ values were 63.9 ± 2.4, 57.8 ± 4.1, 47.8 ± 7.6 μg/ml for HT-29 cells and 57.7 ± 6.6, 47.2 ± 5.2, 34 ± 2.9 μg/ml for HCT116 cells at 24, 48, and 72 h, respectively. Interestingly, FDNVs were not toxic to a normal colon epithelial cell line, CCD 841 CoN. FDNVs exhibited selective uptake by the colorectal cancer cell lines but not the normal colon epithelial cell line. Moreover, dose- and time-dependent FDNV-induced apoptosis was only observed in the colorectal cancer cell lines. In addition, reactive oxygen species levels were substantially increased in colorectal cancer cells, but total glutathione decreased after treatment with FDNVs. Our results show that FDNVs exhibited selective anticancer activity in colorectal cancer cell lines via the disruption of intracellular redox homeostasis and induction of apoptosis, suggesting the utility of FDNVs as a novel intervention for colorectal cancer patients.

## Introduction

Colorectal cancer (CRC) is the third leading cause of cancer-related death and the fourth most frequent malignant tumor worldwide [1]. Several chemotherapeutic drugs are available for

**Funding:** This research project is supported by Mahidol University (Basic Research Fund: fiscal year 2021), Faculty of Science, Mahidol University, the Central Instrument Facility (CIF) Grant, Faculty of Science, Mahidol University and partially supported by Postdoctoral fellowship award from Mahidol University (grant number MD-PD_2021_12). The funders had no role in study design, data collection and analysis, decision to publish, or preparation of the manuscript.

**Competing interests:** The authors have declared that no competing interests exist.

CRC; however, the systemic toxicity to normal cells limits their therapeutic efficacy. These harmful side effects to healthy tissues can be fatal. Therefore, the development of new anticancer agents with fewer toxic side effects is strongly needed [2]. For several decades, traditional medicines from plant extracts and natural compounds have been utilized in cancer treatment [3–6]. *Boesenbergia rotunda* (L.) Mansf., or fingerroot, an herb in the Zingiberaceae family, is a widely found ginger plant in Southeast Asia [7]. Pinostrobin, pinocembrin, and panduratin A are three pharmaceutical bioactive flavonoids isolated from fingerroot [4]. Both extracts and isolated compounds of fingerroot have been found to have anticancer properties in various cancer cells [8, 9]. For example, the fingerroot crude extracts can suppress the growth of nasopharyngeal carcinoma cells (HK1) [8], human promyelocytic cancer cells (HL-60) [10], and human colorectal adenocarcinoma cells (HT-29) [11]. In addition, several isolated compounds from fingerroot have also been reported to have anticancer activities against various cancer cell lines, including human prostate adenocarcinoma (PC3) [12], human lung adenocarcinoma (A549) [9], and human breast cancer (MCF-7) [13]. However, similar to other anticancer compounds, fingerroot extracts and compounds have non-specific cytotoxic effects on non-cancerous cells, thereby limiting their clinical applications [9, 11, 12]. Thus, the efficacy of fingerroot as an anticancer agent is still in question.

In recent years, increasing evidence has shown health benefits of plant-derived nanovesicles (PDNVs). PDNVs are nano-sized, membrane-bound vesicles [14] which contain several biomolecules, including proteins, lipids, mRNAs, and microRNA [15]. Several studies have elucidated the role of PDNVs in intercellular communications through the transferring their components to target recipient cells [16]. An *in vivo* study found that PDNVs can deliver cargo to distant organs via blood circulation and regulate organ function [17]. Furthermore, PDNVs are stable under the acidic conditions of the digestive tract [18]. For example, curcumin encapsulated in PDNVs is four times more stable than free curcumin, leading to efficient intestinal cell absorption [17]. Additionally, oral intake of ginger-derived nanovesicles can help maintain intestinal homeostasis in mice [19]. Moreover, PDNVs have several unique benefits, including lower toxicity, non-immunogenicity, effective target cell uptake, and the ability for large-scale preparation [20, 21]. Thus, PDNVs are emerging as an important factor for therapeutics and targeted drug delivery [21].

Although the anticancer activity of crude extracts and isolated compounds of fingerroot in CRC have been extensively reported, the effect of fingerroot-derived nanovesicles (FDNVs) in CRC is still unknown. Therefore, in the present study, we focused on developing a novel biotherapeutic from fingerroot that selectively targets cancer cells. Specifically, we isolated FDNVs and characterized their properties and therapeutic potential in CRC.

## Materials and methods

### Isolation and purification of fingerroot-derived nanovesicles (FDNVs) and fingerroot extract

Fingerroot was obtained from the Nakhon Pathom province, Thailand. As previously described, isolation of PDNVs was performed with some modifications [22]. The fingerroot was washed 5 times and blended using a clean blender without adding other liquid for homogenization. The blended juice was passed through a cheesecloth to exclude non-homogenized residues and centrifuged twice for 1 h at 10,000×*g* at 4°C. Next, the supernatant was centrifuged again at 50,000×*g* at 4°C for 1 h followed by filtration with a 1.2 μm filter (Acrodisc®, Port Washington, NY, USA). The filtrate was further centrifuged at 100,000×*g* using a fixed-angle rotor 50.2T-Optima L100-XP (Beckman Coulter, Brea, CA, USA) at 4°C for 1.5 h. Next, FDNV pellets were re-suspended in phosphate-buffered saline (PBS), pH 7.4 before filtration

with a 0.45 μm filter (Acrodisc®). The FDNVs were then purified using qEV original size exclusion chromatography (SEC) column (Izon Science, Christchurch, New Zealand) according to the manufacturer's protocol. Thirty fractions were collected.

The fingerroot extract was isolated from *Boesenbergia rotunda* (L.) Mansf. as previously described [23] and kindly provided by Patoomratana Tuchinda, Excellence Center for Drug Discovery (ECDD) and Department of Chemistry, Faculty of Science, Mahidol University.

## Determinations of protein concentration and size distribution of FDNVs

Total protein concentration was measured by Pierce™ BCA Protein Assay Kit (Thermo Fisher Scientific, Waltham, MA, USA) according to the manufacturer's instruction. The size and number of vesicles of fractions 7–9 were determined by nanoparticle tracking analysis (NTA) (Malvern Panalytical, Worcestershire, UK). In brief, 2.5 μl sample was diluted with 1 ml PBS, pH 7.4 to obtain optimal signal count per frame according to the manufacturer's recommendations (30–50 reads/frame). Samples were injected under constant flow conditions at 25˚C, and 3 × 60 s videos were captured. Data were analyzed using NTA 3.4 Build 3.4.003 (Malvern Panalytical).

## Zeta potential measurements

To evaluate FDNVs stability, zeta potential analysis was performed as previously described [24]. Briefly, 50 μl FDNVs were diluted in 1 ml sterile distilled water and applied to a Malvern Zetasizer Nano-ZS ZEN3600 (Malvern Panalytical). Zeta potential measurements were carried out using standard settings (viscosity = 0.89, dielectric constant = 80, temperature = 25˚C). The data were analyzed by the Zetasizer software version 7.11 (Malvern Panalytical).

## Transmission Electron Microscopy (TEM)

Morphology of FDNVs was examined using the negative staining methods [22, 25]. Briefly, drops of FDNVs were deposited onto the surface of a carbon grid and stained with 1% uranyl acetate for 1 min. Images were observed by JEM-1400 TEM (JEOL, Tokyo, Japan) at 100,000X and 300,000X.

## Metabolomic profiling of FDNVs

The metabolites were extracted using the previous protocol with minor modifications [26]. Briefly, FDNVs samples were mixed with methanol and incubated at -20˚C for 48 h. Then, the solution was centrifuged at 15,000·$g$ for 30 min at 4˚C, cleaned using Sep-Pak® C18 Cartridges (Water™, Milford, MA, USA), and vacuum evaporated using a Rotavapor® R-300 (BUCHI, Flawil, Switzerland). The sample was reconstituted in methanol and diluted with 1% formic acid/water at a 1:10 ratio (v/v). Liquid chromatography-mass spectrometry analysis was performed using a Q-Exactive Quadrupole Orbitrap Mass Spectrometer (Thermo Fisher Scientific) coupled to UltiMate 3000 HPLC (Thermo Fisher Scientific). The sample (5 μl) was separated using a Hypersil GOLD™ C18 (Thermo Fisher Scientific) at 28˚C (flow rate of 0.3 ml/min). The total time for each analysis was 35 min. MS was operated in positive mode. A spray voltage of 4.0 kV in both positive, sheath gas and the auxiliary gas flow rate was set at 48 and 11 arbitrary units, respectively. The capillary temperature was 350˚C. The MS analysis alternated between MS full scans and data-dependent MS/MS scans with dynamic exclusion. LC-MS for full MS: scan range, 90–900 m/z; resolution 120,000; AGC target 3e$^6$; max IT 60 ms and LC-MS for full MS/MS: resolution 30,000; AGC target 1e$^5$; max IT 200 ms.

Next, the total ion chromatograms of all the samples were extracted. The acquired raw MS files were processed with Compound Discoverer 3.1 (Thermo Fisher Scientific). The retention time (RT) and mass-to-charge ratio ($m/z$) of different injections were conducted according to the retention time deviation of 0.5 min and the mass deviation of 10 ppm. Then, the peak extraction was performed according to the set information and adduct information: mass deviation = 5 ppm, signal strength deviation = 30%, and signal-to-noise ratio = 2. The target $m/z$ ions were then integrated to predict the molecular formula and compared against mzCloud and ChemSpider online databases to identify and confirm the compounds. Finally, the classes of plant metabolites in FDNVs were classified according to their chemical structure as previously described [27].

## Cell culture

Colorectal cancer (HT-29 and HCT116) and normal human colon epithelial (CCD 841 CoN) cell lines were obtained from the American Type Culture Collection (ATCC, Manassas, VA, USA). HT-29 cells were cultured with Dulbecco's Modified Eagle Medium/Nutrient Mixture F-12 (DMEM/F-12). HCT116 cells were cultured with DMEM low glucose. The medium was supplemented with 10% fetal bovine serum (FBS) and 1% antibiotic-antimycotic (Thermo Fisher Scientific). CCD 841 CoN cells were grown in Eagle's Minimum Essential Medium (EMEM) supplemented with 10% FBS and 1% antibiotic-antimycotic. Cells were maintained in a humidified incubator with 95% $O_2$ and 5% $CO_2$ atmosphere at 37°C. The cells were sub-cultured using 0.05% Trypsin-EDTA (Thermo Fisher Scientific).

## Cytotoxicity assay

Cells were plated at a density of $6.0 \times 10^3$ cells/well in Costar® 96-well plates (Corning Inc., Corning, NY) and grown overnight. Cells were incubated with 3.13 to 100 μg/ml of FDNVs or fingerroot extracts for 24, 48, and 72 h at 37 °C in a humidified 5% $CO_2$ incubator. Untreated cells were used as a negative control. Cell viability was determined by MTT assay (Sigma-Aldrich, St. Louis, MO). Briefly, cells were incubated with 0.5 mg/ml MTT solution at 37°C in a humidified 5% $CO_2$ incubator for 4 h. The medium was then removed before adding 100% DMSO (Sigma-Aldrich). The absorbance was measured at optical density 570 nm using Multiskan™ GO Microplate Spectrophotometer (Thermo Fisher Scientific).

## Apoptosis assay

Cells were seeded at a density of $1 \times 10^5$ cells/well in 24-well plates. After 24 h, cells were treated with FDNVs at concentrations of 25, 50, and 100 μg/ml for 48 h at 37 °C in a 5% $CO_2$ atmosphere. Cells treated with 5% DMSO (Sigma-Aldrich) were used as a positive control. The untreated group was treated equally PBS volume to the treated group. After treatments, cells were washed, detached by trypsin-EDTA, and stained with FITC/Annexin V and propidium iodide (PI) using Annexin V-FITC Apoptosis Detection Kit (BioLegend Way, San Diego, CA, USA) according to the manufacturer's instructions. The stained cells were analyzed using a BD FACSCanto™ flow cytometer (BD Biosciences, San Jose, CA, USA). The data were analyzed by Kaluza Analysis Software version 2.2.1 (Beckman Coulter).

## Quantitative real-time PCR

The expressions of apoptosis-associated genes in FDNVs-treated cells were investigated by quantitative real-time PCR. Briefly, cells ($2 \times 10^5$ cells/well) were seeded in a 12-well plate and treated with 6.25–25 μg/ml FDNVs for 24 h. Total RNAs were extracted using TRIzol™ reagent

(Invitrogen™, Waltham, MA, USA) according to the manufacturer's protocol. cDNA synthesis was conducted using iScript™ Reverse Transcription Supermix (Bio-Rad, Hercules, CA, USA). Targeted gene expressions were determined using iTaq™ Universal SYBR® Green Supermix (Bio-Rad) with specific primers. The expression was normalized to the constitutive expression of GADPH and was calculated using the comparative $2^{-\Delta\Delta CT}$ method [28]. The result is expressed as fold change from three independent experiments carried out in triplicate. Oligonucleotides for the specific primers are as follows: Bax sense strand, 5'-AAGAAGCTGAGC GAGTGT-3' and antisense strand 5'-GGAGGAAGTCCAATGTC-3' [29]; Bcl-2 sense strand, 5'-CTTCTCCCGCCGCTAC-3' and antisense strand 5'-CTGGGGCCGTACAGTTC-3' [29]; Caspase-3 sense strand, 5'-TGCCGTGGTACAGAAC-3' and antisense strand 5'-GAC TCAAATTCTGTTGCC-3' [29]; Caspase-9 sense strand, 5'-CCAGAGATTCGCAAACCA-3' and antisense strand 5'-CCTGACAGCCGTGAGAG-3' [29]; and GAPDH sense strand, 5'-ATGGGGAAGGTGAAGGTCG-3' and antisense strand 5'-GGGTCATTGATGGCAACAA TAT-3' [30].

## Cellular uptake of FDNVs

FDNVs (12.5 and 25 µg/ml) were stained with PKH67 Green Fluorescent Cell Linker Kit (Sigma-Aldrich) according to the manufacturer's instructions. Cells were seeded at a density of $5\times10^4$ cells/well on coverslips in 24-well plates and cultured overnight. Cells were then incubated with PKH67-labeled FDNVs for 24 h at 37 ˚C in a 5% $CO_2$ atmosphere. Non-treated cells were used as a negative control. After incubation, cells were fixed with 4% paraformaldehyde for 20 min and permeabilized with 0.2% Triton X-100 for 10 min at room temperature. Next, nuclei and actin filaments were stained for 30 min with 4′,6-diamidino-2-phenylindole (DAPI) (Invitrogen™) and Alexa Fluor® 647 Phalloidin (Invitrogen™), respectively. Cells were mounted and imaged using an FV1000 confocal laser scanning microscope (Olympus Corporation, Shinjuku, Japan). The fluorescent intensity was quantified using ImageJ software version 1.48h3 (National Institutes of Health; NIH, Bethesda, MD, USA). The mean fluorescence intensity was normalized by the cell number (50,000 cells).

## Inhibition of the cellular uptake of FDNVs

For pinocytosis, cells ($5\times10^4$ cells/well) were pretreated with pinocytosis inhibitors including 1 µg/ml amiloride (macropinocytosis) (Sigma-Aldrich), 5 µg/ml chlorpromazine (clathrin-mediated endocytosis) (Sigma-Aldrich), and 0.25 µg/ml filipin (caveolae-mediated endocytosis) (Sigma-Aldrich) for 1 h [16]. The medium was then removed, and cells were further incubated with 50 µg/ml PKH67-labeled FDNVs in the presence of these inhibitors for 3 h. For phagocytosis, cells were preincubated with 0.005 µg/ml of cytochalasin D (Tocris Bioscience, Bristol, United Kingdom) for 1 h. The medium was then removed and further incubated with FDNVs for 2 h in the presence of this inhibitor. After incubation, cells were fixed, stained with DAPI, and observed using an FV1000 confocal laser scanning microscope (Olympus).

## Measurements of intracellular reactive oxygen species (ROS) levels

Cells were seeded at a density of $1\times10^4$ cells/well in CellCarrier-96 Ultra Microplates (PerkinElmer, Waltham, MA, USA) and incubated overnight. FDNVs were added to the cells at concentrations of 12.5, 25, and 50 µg/ml. After 6 h, cells were washed with Dulbecco's Phosphate Buffered Saline (D-PBS) (Sigma-Aldrich) and incubated with 10 µM CM-H$_2$DCFDA (Thermo Fisher Scientific) at 37 ˚C for 30 min in the dark. Cells incubated with 200 µM $H_2O_2$ (Merck, Darmstadt, Germany) for 3 h were used as positive controls. Cells were washed with D-PBS

and the fluorescence signals were measured using an EnVision® multimode plate reader (PerkinElmer). The level of intracellular ROS was expressed as a ratio to untreated cells.

## Measurements of intracellular glutathione (GSH) levels

Cells were seeded at a density of $2\times10^5$ cells/well in 12-well plates and incubated overnight. Cells were then incubated with FDNVs at concentrations of 12.5, 25, and 50 μg/ml for 6 h at 37 ˚C in a 5% $CO_2$ atmosphere. Cells were then deproteinized with 5% 5-sulfosalicylic acid (Sigma-Aldrich) and centrifuged at 10,000×$g$ at 4˚C for 15 min. Total glutathione levels in the supernatant were measured using a Glutathione Assay Kit (Sigma-Aldrich) according to the manufacturer's instructions.

## Statistical analysis

Data are presented as means ± standard deviation (SD). Statistically significant differences were analyzed by one-way ANOVA and Tukey's multiple comparisons test using GraphPad Prizm software (version 9.0). $P$-value $<0.05$ was considered statistically significant.

## Results

### Isolation, characterization, and metabolite profiling of FDNVs

FDNVs were isolated from fingerroot juice using differential centrifugation, followed by IZON's qEV size exclusion chromatography (SEC) column. Thirty fractions (total volume of 500 μl) were collected from the qEV column, and protein concentrations were determined. We detected a substantial concentration of proteins in fractions 7, 8, and 9 (S1 Fig), with the highest protein concentration observed in fraction 8. Fractions 7–9 contained a high concentration of nanovesicles with high purity. Therefore, the particle number and size distribution of fractions 7, 8, and 9 were determined by nanoparticle tracking analysis. As shown in Fig 1A and Table 1, the range of FDNVs sizes was similar in these 3 fractions, the maximum of which is less than 500 nm. The modal sizes of fractions 7–9 were 78.4 ± 7.8, 70 ± 6.3, and 71.1 ± 1.4, respectively, whereas average particle size of fractions 7–9 were 102.1 ± 4.3, 100.2 ± 10.1, and 106.7 ± 2.4 nm, respectively. Moreover, fraction 8 contained the highest number of particles ($1.5\times10^{11} \pm 7.3\times10^9$ particles/ml) compared to the other fractions (Fig 1B). FDNV morphology was examined by transmission electron microscopy (TEM). The FDNVs were round-shaped membrane-bound vesicles less than 100 nm in size (Fig 1C). Consistent with the NTA result, TEM revealed a greater number of particles in fraction 8 than fractions 7 and 9. In addition, all FDNVs fractions showed negative zeta potential (Fig 1D). Fraction 8 showed the highest negative zeta potential value at -26.9 ± 6.1 mV, whereas fractions 7 and 9 were -17.4 ± 3.7 mV and -10.6 ± 5.6 mV, respectively. This result indicates that fraction 8 showed the highest mutual repulsion and no tendency toward aggregated states. Taken together, the characteristics of our isolated FDNVs are compatible with the previous reports on nanovesicles from edible plants [22, 31]. Therefore, fraction 8 was selected for subsequent experiments.

Next, we investigated the metabolite profiling of FDNVs. As shown in S1 Table, we identified 58 putative metabolites in FDNVs. The distribution of FDNVs metabolites is shown in Fig 1E. Alkaloids were the most common FDNVs metabolites subtype (53%; 31/58), followed by phenolics (21%; 12/58), lipids (14%; 8/58), and organic compounds (12%; 7/58). Importantly, the phenolic compounds naringenin chalcone, pinostrobin, and pinocembrin were found in FDNVs. These phenolic compounds have been found as promising bioactive compounds in fingerroot juice [4].

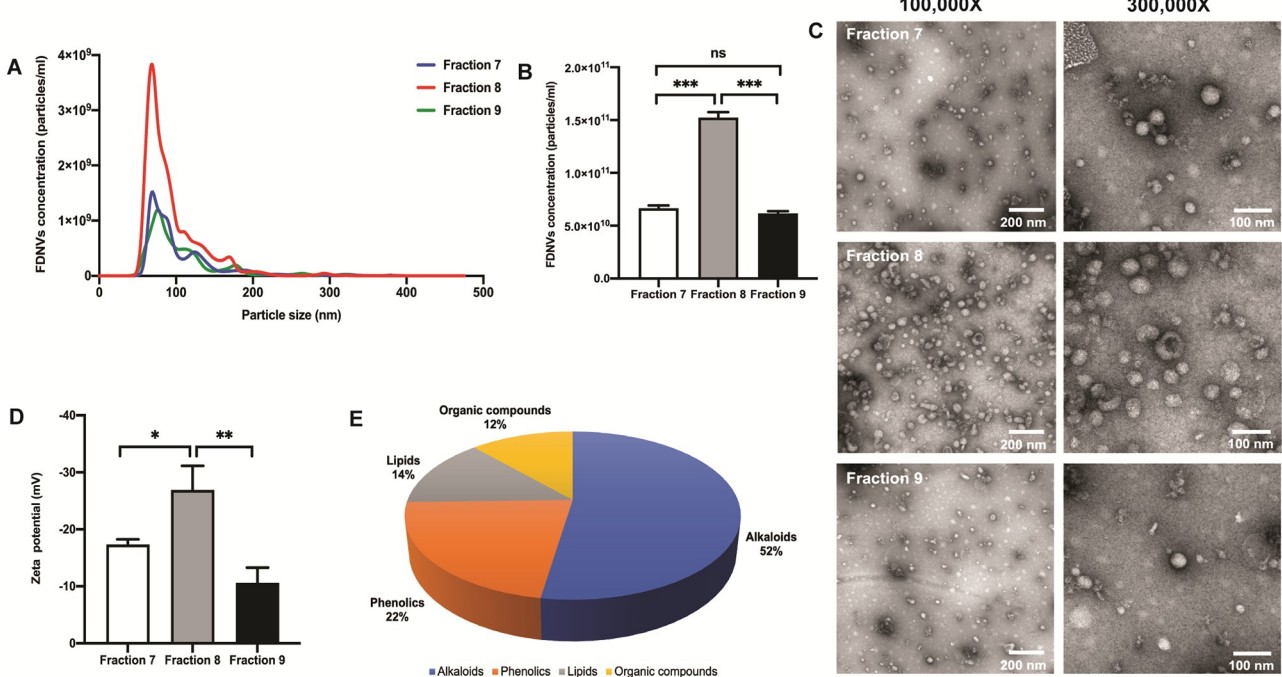

**Fig 1. Characterization of FDNVs.** (A) The size distribution and (B) particle concentration of fractions 7, 8, and 9 were analyzed by NTA. (C) FDNV morphology was observed under TEM at 100,000X (scale bar = 200 nm) and 300,000X (scale bar = 100 nm). (D) The zeta potential of isolated FDNVs was measured using a Zetasizer. (E) Distribution of tentatively identified metabolites in FDNVs using LC-MS/MS. Data are represented as means ± SD of three independent experiments in duplicate; ns = not significant. $^*P < 0.05$, $^{**}P < 0.01$, $^{***}P < 0.001$ (one-way ANOVA).

## Cytotoxicity of FDNVs and fingerroot extract on colorectal cancer cells

We next investigated the cytotoxic effect of FDNVs against two colorectal cancer cell lines, HT-29 and HCT116. FDNVs exhibited dose- and time-dependent cytotoxic effects against both CRC cell lines (Fig 2A and 2B and Table 2). At 25 µg/ml, FDNVs caused cytotoxicity on HT-29 and HCT116 cells after 24 h of incubation. The IC$_{50}$ values of FDNVs against HT-29 cells were 63.9 ± 2.4, 57.8 ± 4.1, and 47.8 ± 7.6 µg/ml at 24, 48, and 72 h, respectively. The IC$_{50}$ values of FDNVs against HCT-116 cells were 57.7 ± 6.6, 47.2 ± 5.2, and 34 ± 2.9 µg/ml at 24, 48, and 72 h, respectively. Interestingly, FDNVs had no cytotoxic effects toward normal human colon epithelial cells (CCD 841 CoN) (Fig 2C). In addition, we compared the cytotoxic selectivity between fingerroot extract and its nanovesicles. In contrast to the selective cytotoxic effect of FDNVs, fingerroot extract exhibited dose- and time-dependent effects against both cancer cells and normal human colon epithelial cells (Fig 2 and Table 2). Cytotoxicity of fingerroot extract was significantly observed at 25 µg/ml after 24 h of treatment toward all tested cells ($P < 0.001$). Additionally, there was no difference between the IC$_{50}$ values of the

**Table 1. Size of FDNVs in fractions 7, 8 and 9 using nanoparticle tracking analysis (n = 3).**

| Fraction | Min size (nm) | Max size (nm) | Modal size (nm) | Mean size (nm) |
|----------|---------------|---------------|-----------------|----------------|
| F7 | 33.5 ± 2.7 | 428.5 ± 25.3 | 78.4 ± 7.8 | 102.1 ± 4.3 |
| F8 | 34 ± 1.1 | 483.5 ± 15.4 | 70 ± 6.3 | 100.2 ± 10.1 |
| F9 | 33.2 ± 9.2 | 419 ± 8.8 | 71.1 ± 1.4 | 106.7 ± 2.4 |

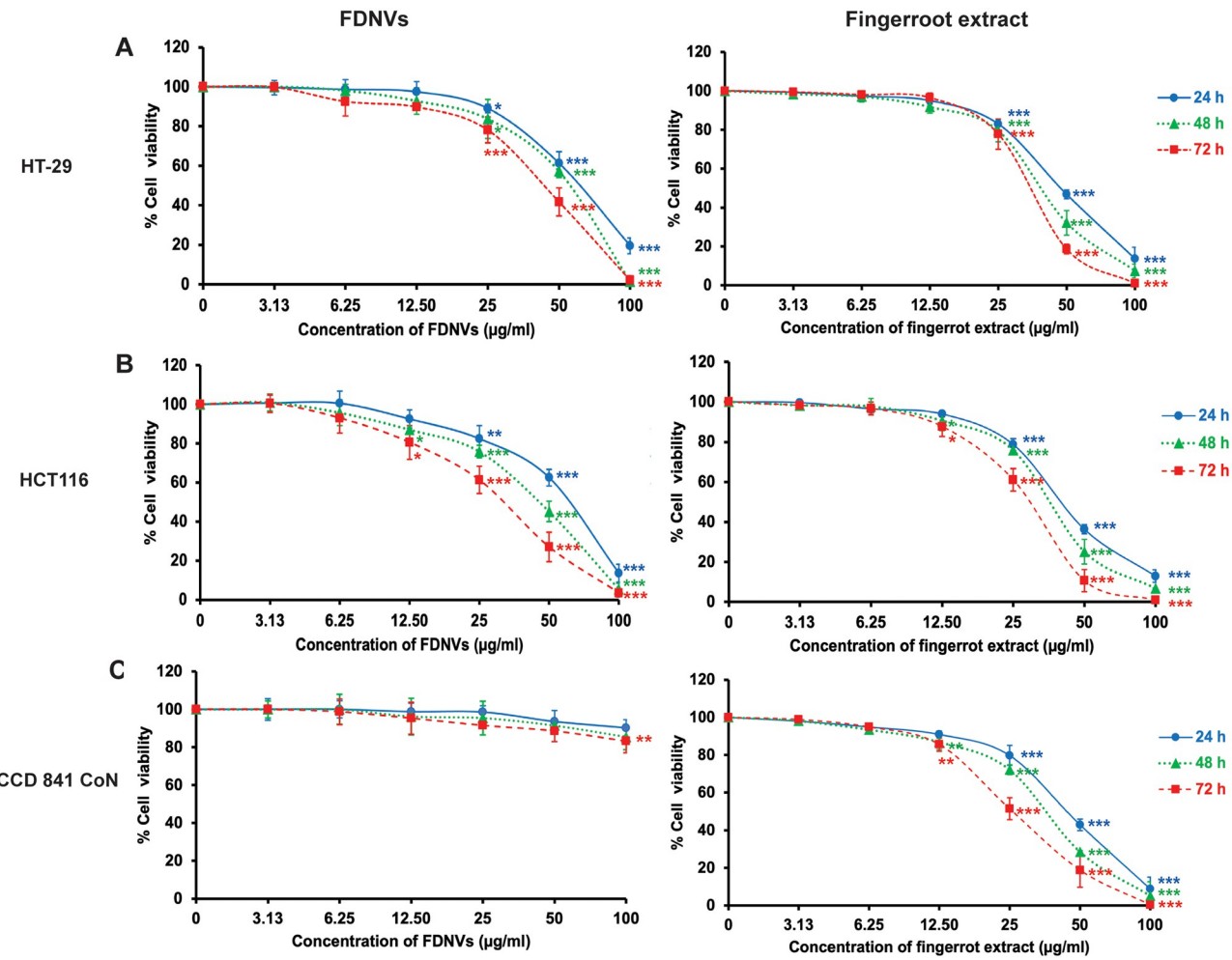

**Fig 2. Cytotoxic effects of FDNVs and fingerroot extract on colorectal cancer and normal colon epithelial cell lines.** Cell viabilities of (A) HT-29, (B) HCT116, and (C) CCD 841 CoN cells were determined using MTT assay after treatment with FDNVs and fingerroot extract for 24, 48, and 72 h. Data are represented as means ± SD of three independent experiments in triplicate. $^*P < 0.05$, $^{**}P < 0.01$, $^{***}P < 0.001$ (one-way ANOVA).

**Table 2. IC$_{50}$ values of FDNVs and fingerroot extract against colorectal cancer cells.**

| Cell line | IC$_{50}$ (µg/ml) | | | | | |
|---|---|---|---|---|---|---|
| | FDNVs | | | Fingerroot extract | | |
| | 24 h | 48 h | 72 h | 24 h | 48 h | 72 h |
| HT-29 | 63.9 ± 2.4 | 57.8 ± 4.1 | 47.8 ± 7.6 | 59.1 ± 0.5 | 46.3 ± 3.5 | 34.9 ± 2.9 |
| HCT116 | 57.7 ± 6.6 | 47.2 ± 5.2 | 34 ± 2.9 | 57.7 ± 3.2 | 40.8 ± 3.3 | 30.8 ± 1.7 |
| CCD 841 CoN | N/A | N/A | N/A | 57.6 ± 2.9 | 44.3 ± 0.3 | 29.6 ± 3.8 |

N/A = Not applicable.

fingerroot extract against all tested cells. These results indicate the selective cytotoxic effect of FDNVs on colorectal cancer cells with relatively low cytotoxicity toward normal colon cells.

## Cellular uptake of FDNVs

To investigate the differential cytotoxic effect of FDNVs on CRC and colon epithelial cells, we next examined the uptake of FDNVs into cancer cells and normal colon epithelial cells (Fig 3). Cells were incubated with 12.5 µg/ml FDNVs labeled with PKH67 (green) for 24 h. The intracellular green fluorescence signals of PKH67-labeled FDNVs were detected in both CRC cell lines (Fig 3A and 3B). Moreover, intracellular fluorescence was positively correlated with FDNV concentration in CRC cells, as we observed increased fluorescence after incubation with at a higher concentration of FDNVs (25 µg/ml). In contrast, we detected significantly lower fluorescence intensity of PKH67-labeled FDNVs in CCD 841 CoN cells (Fig 3C) compared with HT-29 ($P < 0.001$) and HCT116 ($P < 0.001$) cells (Fig 3A, 3B and 3D). There were no green fluorescence signals in vehicle control-treated cells. The quantification of fluorescence intensity per 50,000 cells is shown in Fig 3D. These findings may partially explain the selective cytotoxic effect of FDNVs on colorectal cancer cells.

Since the selective cytotoxic effects of FDNVs likely involve cellular uptake, we investigated the uptake mechanism of FDNVs in colorectal cells. Generally, there are two major

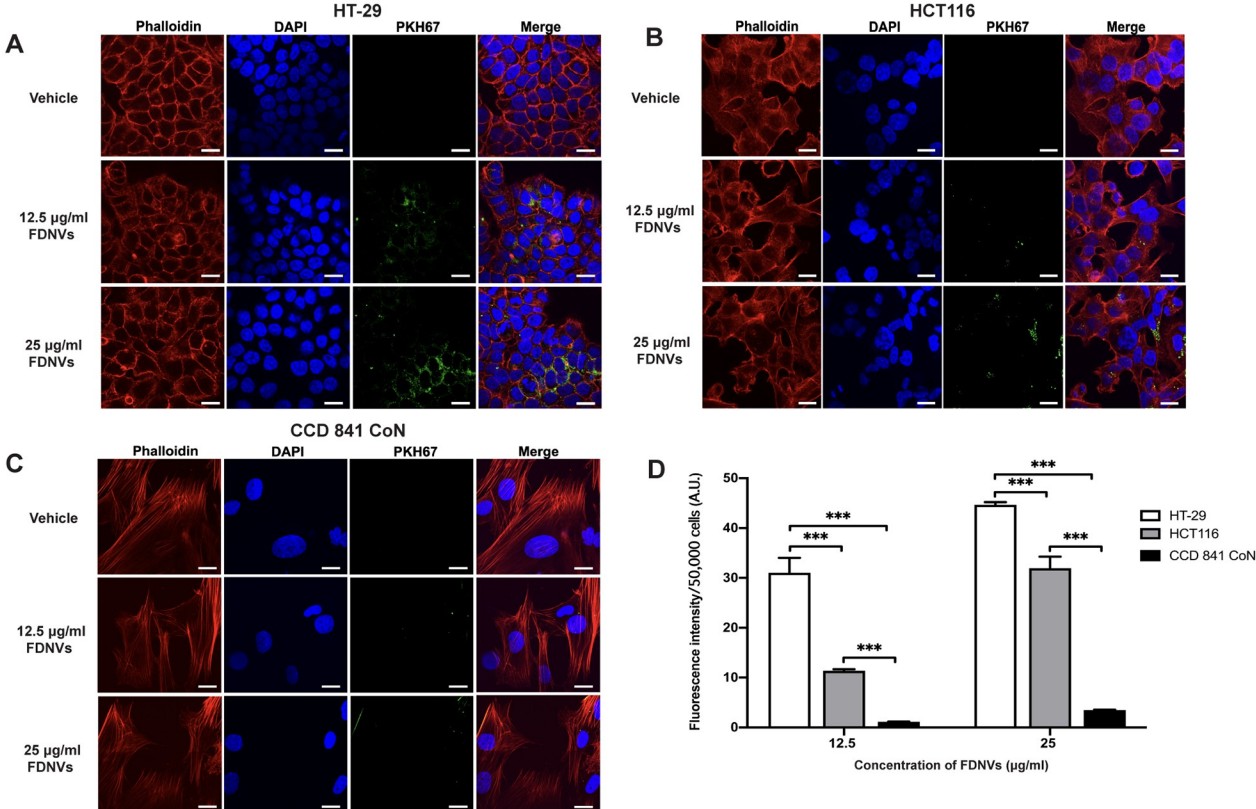

**Fig 3. Internalization of FDNVs in colorectal cancer and normal colon epithelial cells.** (A) HT-29, (B) HCT116, and (C) CCD 841 CoN cells were incubated with PKH67-labeled FDNVs (green) for 24 h. Cells were also stained with DAPI (blue) and Phalloidin (red) to label the nucleus and actin filaments, respectively. (D) The green fluorescence intensity of FDNVs was determined using ImageJ software. The values are shown as means ± SD of three independent experiments in duplicate. ***$P < 0.001$ (one-way ANOVA). Scale bar = 10 µm.

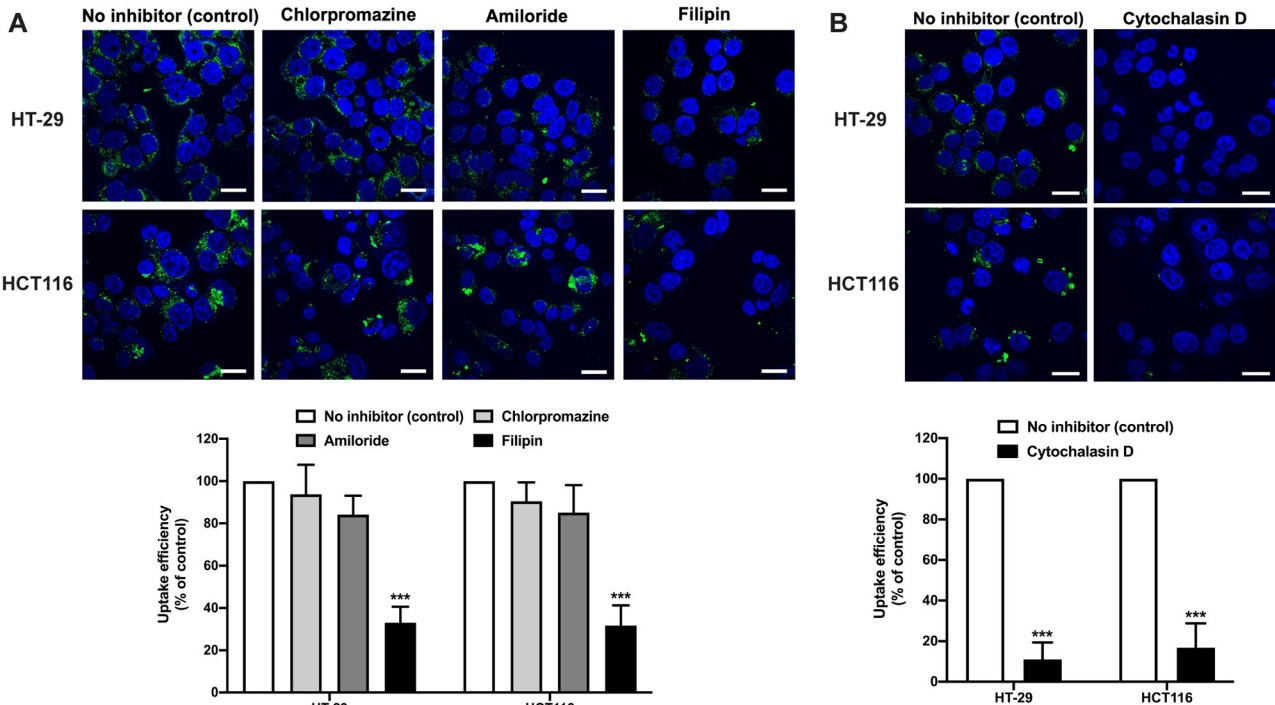

**Fig 4. The inhibition of FDNVs internalization in colorectal cells.** (A) HT-29 and HCT116 cells were pretreated with chlorpromazine, amiloride, and filipin for 1 h and then incubated with PKH67-label FDNVs (green) for an additional 3 h in the presence of the inhibitors. (B) HT-29 and HCT116 cells were pretreated with cytochalasin D for 1 h and then incubated with PKH67-label FDNVs for an additional 2 h in the presence of the inhibitor. Cells were then fixed and stained with DAPI (blue). The green fluorescence intensity of FDNVs was determined by ImageJ software. Bar graphs show the uptake efficiency of FDNVs in HT-29 and HCT116 cells. The values are presented as means ± SD of three independent experiments in duplicate. ***$P < 0.001$ (one-way ANOVA). Scale bar = 10 μm.

endocytosis pathways, including phagocytosis and pinocytosis. Pinocytosis divides into three subcategories: micropinocytosis, clathrin-mediated endocytosis, and caveolae-mediated endocytosis [16]. Thus, we incubated cells with FDNVs and uptake inhibitors to block cellular uptake. As shown in Fig 4, the uptake of FDNVs in HT-29 and HCT116 were markedly inhibited by filipin (Fig 4A) and cytochalasin D (Fig 4B), which are inhibitors of caveolae-mediated endocytosis and phagocytosis, respectively ($P > 0.001$). Conversely, treatment with amiloride, an inhibitor of micropinocytosis, and chlorpromazine, an inhibitor of clathrin-mediated endocytosis, did not affect the uptake of FDNVs in both cancer cell lines (Fig 4A). These data suggest that the internalization of FDNVs in colorectal cancer cells is partly via caveolae-mediated endocytosis and phagocytosis pathways.

## FDNVs induce colorectal cancer cells apoptosis

The effect of FDNVs on apoptosis induction in CRC cells was further investigated by FITC/ Annexin V staining and flow cytometry (Fig 5). As shown in Fig 5A and 5B, treatment with FDNVs markedly induced apoptosis in HT-29 and HCT116 cells in a dose-dependent manner. Treatments with FDNVs at 25, 50, and 100 μg/ml significantly caused early apoptosis in HT-29 cells, up to 6.4 ± 1.2% ($P < 0.05$), 12.2 ± 2.4% ($P < 0.001$), and 18.1 ± 2.4% ($P < 0.001$), respectively, compared to untreated control (2.9 ± 1.1%) (Fig 5A). In addition, late apoptotic population increased in these cells after treatment with FDNVs at 50 and 100 μg/ml compared with control. Similarly, treatment with FDNVs at 50 and 100 μg/ml significantly induced early

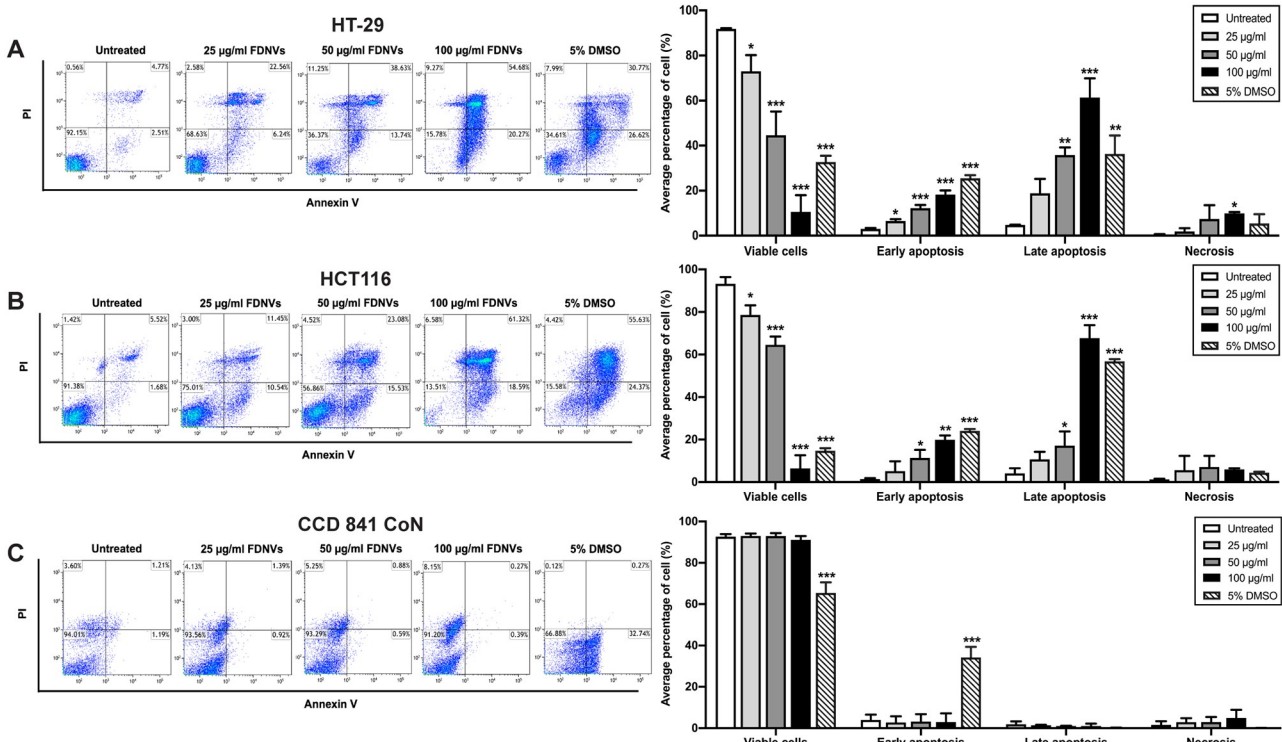

**Fig 5. Representative FACS quantitative analyses showing FDNVs-induced apoptosis.** (A) HT-29, (B) HCT116, and (C) CCD 841 CoN cells were treated with FDNVs at indicated concentrations for 48 h. The apoptosis induction of FDNVs-treated cells was examined using flow cytometry-based Annexin V staining. Data were analyzed using Kaluza analysis software and shown as means ± SD of three independent experiments in duplicate. $^{*}P < 0.05$, $^{**}P < 0.01$ and $^{***}P < 0.001$ (one-way ANOVA).

apoptosis in HCT116 cells, up to 11.3 ± 5.6% ($P < 0.05$) and 19.8 ± 3.1% ($P < 0.01$), respectively, compared to untreated cells (1.4 ± 0.7%). Late apoptotic population also increased in HCT116 cells treated with 50 and 100 µg/ml FDNVs (Fig 5B). However, a significant increase in necrotic cell death was found only in the HT-29 cells after treatment with a higher dose of 100 µg/ml FDNVs ($P < 0.05$) (Fig 5A). In addition, the percentage of viable cells decreased in both CRC cell lines after treatment with FDNVs (Fig 5A and 5B). These results indicate that the cytotoxic effect of FDNVs in CRC cells are mediated through apoptosis induction. To further elucidate the mechanism of the differential cytotoxic effects of FDNVs on CRC and human colon epithelial cells, we examined FDNV-induced apoptosis in normal human colon epithelial cells, CCD 841 CoN (Fig 5C). There was no significant induction of early apoptosis in all tested concentrations of FDNVs, as compared to untreated control. Statistically significant induction of apoptosis was only found in the presence of 5% DMSO in human colon epithelial cells (32.2 ± 5.1%, $P < 0.001$). More than 90% of cells remained viable even at a high concentration of FDNVs, indicating that FDNVs exhibited low cytotoxicity against normal colon cells. In contrast, treatment with 5% DMSO resulted in a significant reduction of cell viability ($P < 0.001$) relative to control. Late apoptosis and necrosis were not significantly different in all tested conditions. These results demonstrated that FDNVs displayed selective induction of apoptosis-mediated cell death in cancerous, but not normal, cells.

To further confirm the underlying mechanism of apoptosis induction of FDNVs, the effect of FDNVs treatment on the expression of apoptosis-related genes was examined by quantitative RT-PCR analysis (Fig 6). Treatment with FDNVs at 12.5 and 25 µg/ml markedly increased

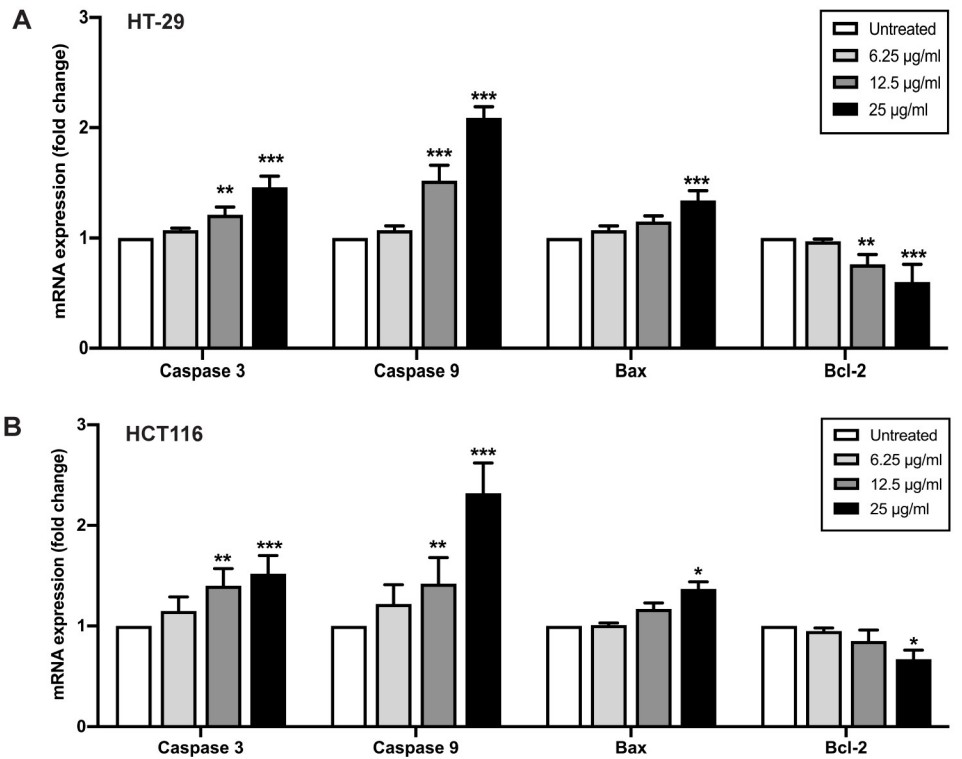

**Fig 6. Effect of FDNVs on apoptosis-related genes expression.** (A) HT-29 and (B) HCT116 cells were treated with FDNVs for 24 h. The expression of target genes was determined using quantitative RT-PCR. The relative quantitation of each gene was normalized to the constitutive expression of GADPH. The results are mean ± SD of three independent experiments in duplicate and presented as fold change. $^*P < 0.05$, $^{**}P < 0.01$ and $^{***}P < 0.001$ compared with untreated cells (one-way ANOVA).

the expression of caspase-3 and caspase-9 in HT-29 and HCT116 cells (Fig 6A and 6B). An increase in the expression of Bax, a pro-apoptotic gene, was also observed after treatment with 25 μg/ml FDNVs in both CRC cell lines. In contrast, the expression of Bcl-2, an anti-apoptotic gene, was decreased in both CRC cell lines after treatment with FDNVs at 25 μg/ml. These results indicate that FDNVs-mediated apoptosis induction in cancer cells is associated with the up-regulation of caspase and pro-apoptotic genes and the suppression of an anti-apoptotic gene.

## FDNVs increased ROS generation but decreased glutathione levels in colorectal cancer cells

We next investigated whether FDNVs-induced apoptosis was mediated by reactive oxygen species (ROS) (Fig 7). Treatment with FDNVs at 12.5, 25, and 50 μg/ml significantly increased ROS levels in HT-29 cells, up to 1.2 ± 0.18 ($P < 0.05$), 1.3 ± 0.04 ($P < 0.01$), and 1.4 ± 0.09 ($P < 0.001$), respectively, compared to untreated control (Fig 7A). A similar result was observed in HCT116 (Fig 7B). Relative to control-treated cells, ROS levels were significantly increased up to 1.5 ± 0.22% ($P < 0.01$), 1.7 ± 0.14% ($P < 0.001$), and 1.8 ± 0.21% ($P < 0.001$) in HCT116 cells treated with FDNVs at 12.5, 25, and 50 μg/ml, respectively. Similarly, treatment with $H_2O_2$, a positive control, significantly increased ROS concentrations up to 1.3 ± 0.13 ($P < 0.01$) in HT-29 cells and 1.4 ± 0.09 ($P < 0.05$) in HCT116 cells. Additionally,

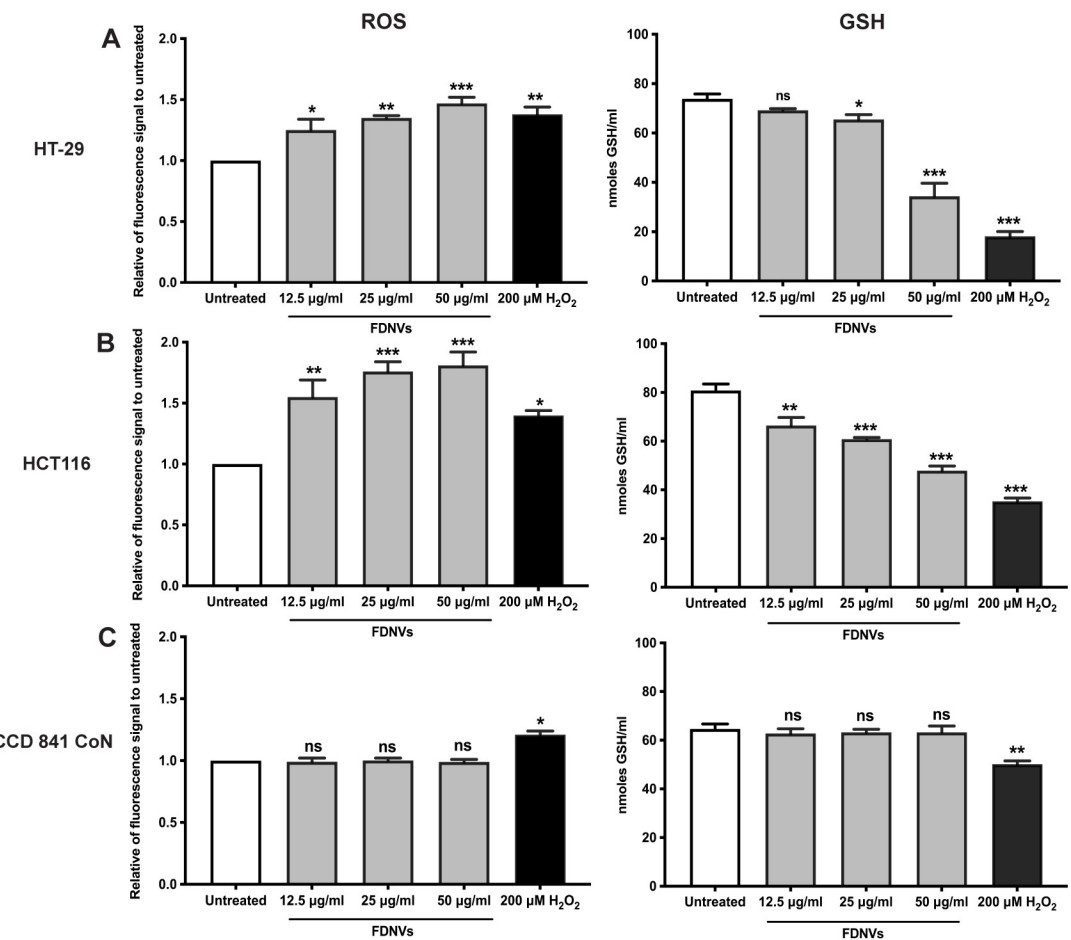

**Fig 7. Induction of intracellular ROS and GSH in FDNVs-treated cells.** (A) HT-29, (B) HCT116, and (C) CCD 841 CoN cells were treated with FDNVs for 6 h. Cells with 200 μM $H_2O_2$ for 3 h were used as a positive control. Intracellular ROS and GSH were determined using CM-$H_2$DCFDA and Glutathione assay kit, respectively. The levels of intracellular ROS are represented as a ratio to untreated control. The levels of intracellular GSH are represented as mean ± SD. All experiments were performed in three independent experiments in duplicate: ns = not significant. $^*P < 0.05$, $^{**}P < 0.01$ and $^{***}P < 0.001$ (one-way ANOVA).

treatment with FDNVs did not affect the induction of intracellular ROS in CCD 841 CoN (Fig 7C).

To examine the disruption of redox balance, we determined the level of glutathione (GSH) in FDNVs-treated cells (Fig 7). Treatments of HT-29 cells with FDNVs at 25 and 50 μg/ml significantly decreased GSH levels to 65.5 ± 3.2 nmoles/ml ($P < 0.05$) and 34.3 ± 7.2 nmoles/ml ($P < 0.001$), respectively, which was significantly lower than control-treated cells (73.8 ± 3.1 nmoles/ml) (Fig 7A). Similarly, after treatment with FDNVs at 12.5, 25, and 50 μg/ml, GSH levels in HCT116 cells were significantly reduced to 66.4 ± 3.9 nmoles/ml ($P < 0.01$), 60.8 ± 1.3 nmoles/ml ($P < 0.001$), and 47.8 ± 2.7 nmoles/ml ($P < 0.001$), respectively. This was significantly lower than untreated cells (80.8 ± 3.9 nmoles/ml) (Fig 7B). Treatment with $H_2O_2$, as a positive control, reduced GSH levels in both CRC cell lines (Fig 7A and 7B). In contrast to CRC cells, in CCD 841 CoN cells, only $H_2O_2$ causes significantly reduced levels of GHS ($P < 0.01$, Fig 7C). These data suggest that FDNVs showed selective cytotoxicity towards cancer cells through increased ROS production. Moreover, FDNVs-induced apoptosis in CRC

cell lines is possibly due to disruption of the redox balance leading to apoptotic cell death; thus, FDNVs have significant potential to be developed as selective anticancer drugs.

## Discussion

In the present study, we provided the first isolation and characterization of nanovesicles from fingerroot. More importantly, we demonstrated promising selective anticancer effects of these nanovesicles against CRC cells. As such, fingerroot-derived nanovesicles (FDNVs) exerted their anticancer activity by stimulating apoptotic mechanisms mediated through ROS production. Furthermore, FDNVs did not cause toxicity to normal human colon epithelial cells. These findings highlight an alternative approach in using nanovesicles from natural sources in cancer therapy.

Fingerroot possess several pharmacological activities, including antiviral [23], anti-inflammatory [32], and potential anticancer [8, 11] effects. Currently, the nanovesicles extracted from plants exhibit excellent potential for therapeutic applications against various diseases [33]. With the recent increased interest in the therapeutic potential of plant-derived nanovesicles (PDNVs), several groups have attempted to isolate and develop plants as natural green nano-factories to investigate their biomedical utility [21, 22]. However, the isolation, characterization, and biological activity of nanovesicles isolated from fingerroot or FDNVs have not been reported. In this study, we established a protocol to isolate nanovesicles from fingerroot. The standard protocol for PDNV isolation and characterization has only recently been fully developed [14]. Differential centrifugation is widely used for nanovesicles isolation. Consequently, size-, density- and immunoaffinity-based techniques have been applied to purify and reduce non-vesicular extracellular materials [34]. Here, we used the differential centrifugation method followed by a qEV size exclusion chromatography (SEC) column to isolate and purify the FDNVs. We found that the isolated FDNVs from our protocol exhibited characteristics of nanovesicles similar to those isolated from other edible plants [22, 31]. The FDNVs were approximately 100 nm in diameter, which was similar to the report of nanovesicles derived from ginger [16]. Although the previous study showed that nanovesicle isolation using the immunoaffinity-based technique (ExoQuick plus) provided the highest particle concentration, the qEV column and sucrose density-gradient separation methods contain less protein contamination than the immunoaffinity-based technique [34, 35]. Therefore, the established protocol for isolation FDNVs in this study may help isolate nanovesicles from other types of plants.

Plant secondary metabolites play a crucial role in the pharmacological actions of medicinal plants [27]. Our FDNVs contained alkaloids, phenolics, lipids, and organic compounds, which illustrates the diversity of phytochemical constituents in FDNVs. There have been reports of anticancer potentials of phenolic compounds via ROS-mediated apoptosis, such as naringenin chalcone [36], pinostrobin [37], and pinocembrin [38]. In addition, valerenic acid (lipid) and darymid A (alkaloid) have also been found to possess anticancer activity [39, 40]. Perhaps these secondary metabolites may serve as medicinal agents that underlie the therapeutic action of FDNVs, which can improve our understanding of how FDNVs exhibit biological activities. Current chemotherapeutic drugs for CRC have off-target effects that cause toxicity to both cancer cells and their normal counterparts [9, 11, 12]. Therefore, finding anticancer agents with a high level of specificity may help reduce side effects for CRC patients and improve their quality of life [2]. Previously, extracellular vesicles from plant-sap have been revealed to have selective cytotoxic effects on tumor cells rather than normal cells [16]. Herein, we demonstrated that FDNVs exhibit anticancer activity against two CRC cell lines (HT-29 and HCT116); however, FDNVs have reduced cytotoxicity toward normal colon epithelial cells

(CCD 841 CoN). On the contrary, the fingerroot extract exhibited a cytotoxic effect against both CRC and normal colon cells. This is consistent with other studies, which have found anticancer activity of fingerroot extract against several cancer cell lines [8, 10, 11]. However, fingerroot extract induced cytotoxicity on non-cancerous cells, such as non-transformed human skin fibroblast cells (SF 3169) [11], normal hepatic cells (WRL68) [12], and normal colon epithelial cells (CCD 841 CoN) [9]. Importantly, our results illustrate a tremendous potential of FDNVs to selectively target CRC cells relative to the parental fingerroot extract.

Our study also found that both types of CRC cells were more susceptible to FDNV uptake than normal colon cells. FDNVs were taken up and preferentially localized in the cytoplasm of cancer cells. For example, ginger-derived nano-lipids loaded with doxorubicin were mainly internalized via the phagocytosis pathway into CRC cancer cells that were significantly inhibited by cytochalasin D [41]. Moreover, the internalization of plant sap-derived extracellular vesicles in breast cancer cells was mediated by phagocytosis and caveolae-mediated endocytosis [16]. Thus, our findings are consistent with other reports that the internalization of FDNVs into CRC cells is likely due to phagocytosis and caveolae-mediated endocytosis. Caveolin-1, the principal structural component of caveolae, is involved in caveolae-mediated endocytosis [42]. Although caveolin-1 function in cancer is controversial, overexpression of caveolin-1 has been reported in colon cancer [43]. Therefore, the caveolae-mediated endocytosis may be more effective in CRC, resulting in larger amounts of FDNVs internalization than in normal colon epithelial cells. In addition, the internalization of garlic-derived nanovesicles is mediated by interaction with the CD98 heavy chain (CD98hc) in liver cancer cells (HepG2) [44]. Expression levels of CD98hc protein were higher in CRC tissues than in matched normal tissues [45]. Therefore, upregulation of CD98hc might support the uptake of FDNVs in CRC cells. Taken together, these specific properties may help cancer to gain nanovesicles uptake inside the cells and explain the greater toxicity of FDNVs toward CRC cells. However, additional experiments are required to understand the FDNVs uptake mechanism in cancer cells.

Our study showed that FDNVs drove apoptosis cell death in CRC cell cultures. The well-known apoptotic mechanism is initiated by the induction of the intrinsic pathway via the targeting and activation of caspase-9 in response to the release of cytochrome c, consequently leading to the activation of effector caspases (-3, -6, and -7) [46]. We detected an increased expression of pro-apoptotic genes, including Caspase-3, Caspase-9, and Bax, as well as a decrease in the expression of the anti-apoptotic gene Bcl-2, in CRC cells after FDNVs treatment. These findings indicate that the anticancer effect of FDNVs was partially mediated through activation of the intrinsic pathway, leading to the execution of apoptosis.

The induction of intracellular ROS reportedly contributes to apoptosis in cancer cells [47, 48]. Several studies have found that medicinal plants can cause excessive production of ROS, leading to irreversible damage to DNA, lipids, and proteins, ultimately leading to the induction of apoptosis [49]. The secondary metabolites present in plants, such as flavonoids, fatty acids, and proteins have been shown to induce ROS generation, i.e. pinostrobin [37], linoleic acid [50], and phospholipase D [51], respectively. Our study showed that ROS levels were elevated in HT-29 and HCT116 cells after FDNVs treatment. Indeed, previous studies have reported ROS induction in response to fingerroot compounds. Boesenbergin A, a chalcone from fingerroot induced oxidative stress-mediate apoptosis in lung adenocarcinoma cells (A549) [9]. Pinostrobin, a flavanone in Fingerroot, exhibited anti-proliferation effects and induced apoptosis in cancer stem-like cells through a ROS-dependent mechanism [37]. Additionally, nanovesicles from several plants, including lemon and ginseng, have also reported ROS-mediated apoptosis [21, 31]. Hence, FDNVs may contain bioactive compounds that play a key role in ROS generation, leading to the induction of apoptosis in CRC cells. Besides the metabolites, plant-derived microRNAs have also been reported to exhibit anticancer effect [52]. Therefore,

the miRNA profile in FDNVs needs further investigation to more directly address the molecular mechanism of FDNVs-mediated anticancer properties. Interference of cellular detoxification by reducing GSH was associated with ROS-mediated cytotoxicity. [47, 53]. We, therefore, determined the cellular level of GSH. Indeed, we found significant reductions of GSH in all treated CRC cells, which supports our hypothesis on the disruption of redox balance in FDNVs-treated cells by reducing GSH to neutralize ROS. On the other hand, the ROS and GSH levels were not significantly altered in normal colon cells after treatment with FDNVs. These findings suggest that FDNVs promoted apoptosis through the production of intracellular ROS and the GSH system's dissipation, resulting in cytotoxicity selectively towards CRC cells.

Recent interest in the study of PDNVs is partly due to their various biological properties, which can have selective targeting, leading to novel opportunities for clinical applications in various diseases [14]. Additional advantages of PDNVs include large-scale production, possessing high biocompatibility, and stability under gastrointestinal tract conditions [33]. However, there remains the absence of a standard protocol of PDNV isolation, as there is presently no consensus among researchers. In addition, specific protein markers for PDNVs are still controversial due to the robust diversity between different species of plants. Hence, the precise understanding of the composition and biological functions of PDNVs may help establish a standard isolation method and improve therapeutic applications.

## Conclusion

In conclusion, this study demonstrated the anticancer effect of FDNVs against CRC cells with low toxicity to normal colon epithelial cells, indicating its selective anticancer property. Furthermore, the anticancer effect of FDNVs was mediated through disruption of intracellular redox homeostasis and induction of apoptosis pathway. Thus, FDNVs may be a promising intervention for CRC patients.

## Supporting information

**S1 Fig. Total protein concentration of samples from qEV column.** Protein concentrations of 30 fractions from qEV were determined by BCA Protein Assay.
(TIF)

**S1 Table. The discriminative putatively identified metabolites of FDNVs.** The metabolites were identified based on ChemSpider online databases, with rigorous statistical validation.
(DOCX)

**S2 Table. The minimal data set underlying the results.**
(DOCX)

## Author Contributions

**Conceptualization:** Saharut Wongkaewkhiaw, Arthit Chairoungdua, Nittaya Boonmuen.

**Data curation:** Saharut Wongkaewkhiaw, Arthit Chairoungdua, Nittaya Boonmuen.

**Formal analysis:** Saharut Wongkaewkhiaw, Arthit Chairoungdua, Nittaya Boonmuen.

**Funding acquisition:** Saharut Wongkaewkhiaw, Nittaya Boonmuen.

**Investigation:** Saharut Wongkaewkhiaw, Arthit Chairoungdua, Nittaya Boonmuen.

**Methodology:** Saharut Wongkaewkhiaw, Amaraporn Wongrakpanich, Sucheewin Krobthong, Arthit Chairoungdua, Nittaya Boonmuen.

**Writing – original draft:** Saharut Wongkaewkhiaw, Arthit Chairoungdua, Nittaya Boonmuen.

**Writing – review & editing:** Saharut Wongkaewkhiaw, Amaraporn Wongrakpanich, Witchuda Saengsawang, Arthit Chairoungdua, Nittaya Boonmuen.

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
