## [Decision Letter · Decision Letter 0]

1 Nov 2021

PONE-D-21-31088Induction of apoptosis in human colorectal cancer cells by nanovesicles from fingerroot (Boesenbergia rotunda (L.) Mansf.)PLOS ONE

Dear Dr. Boonmuen,

Thank you for submitting your manuscript to PLOS ONE. After careful consideration, we feel that it has merit but does not fully meet PLOS ONE’s publication criteria as it currently stands. Therefore, we invite you to submit a revised version of the manuscript that addresses the points raised during the review process.

Please carefully read all the comments by the reviewers and address them point by pointPlease indicate the changes in the manuscriptPlease address all typographical and grammatical errors in the manuscript==============================

We look forward to receiving your revised manuscript.

Kind regards,

Lay-Hong Chuah

Academic Editor

PLOS ONE

Journal Requirements:

“This research project is supported by Mahidol University (Basic Research Fund: fiscal year 2021), Faculty of Science, Mahidol University, the Central Instrument Facility (CIF) Grant, Faculty of Science, Mahidol University and partially supported by Postdoctoral fellowship award from Mahidol University (grant number MD-PD_2021_12).”

Additional Editor Comments:

Please address all comments by the reviewers and resubmit.

Reviewers' comments:

Reviewer's Responses to Questions

**Comments to the Author**

1. Is the manuscript technically sound, and do the data support the conclusions?

Reviewer #1: No

Reviewer #2: Yes

Reviewer #3: Yes

Reviewer #4: Partly

2. Has the statistical analysis been performed appropriately and rigorously? 

Reviewer #1: Yes

Reviewer #2: Yes

Reviewer #3: Yes

Reviewer #4: Yes

3. Have the authors made all data underlying the findings in their manuscript fully available?

Reviewer #1: Yes

Reviewer #2: Yes

Reviewer #3: Yes

Reviewer #4: Yes

4. Is the manuscript presented in an intelligible fashion and written in standard English?

Reviewer #1: No

Reviewer #2: Yes

Reviewer #3: Yes

Reviewer #4: No

5. Review Comments to the Author

Reviewer #1: In this manuscript, the authors report the nanovesicles derived from fingerroot possess the anticancer activity against CRC cell lines (HT-29 and HCT116) and are inert to the normal colon epithelial cells. But this manuscript is too preliminary in science. I suggest the authors to analyze the fingerroot derived nanovesicle components by metabolomics, proteomics, and transcriptomics, and investigate the cellular internalization mechanism of fingerroot derived nanovesicles. A better understanding of fingerroot derived nanovesicle component and cellular uptake is essential to potentiate the capacity of nanovesicles to induce phenotypic changes in recipient cells. The introduction and discussion sections in this manuscript are also poorly constructed and written. Although their approach itself is of interest, the presentation does not suffice to show the novelty and significance that meet the standards of rigor required by the journal to be considered for publication.

Reviewer #2: The present work is interesting as authors isolated EVs from fingerroot for the treatment of CRC. I have carefully reviewed the articles and my comments are as follows:

Introduction:

1) Line 47-48. The unsatisfactory response rate is referring to all the drugs used for colon cancer? Early or late stage or patient? This has to be clear otherwise it shows confusing statement. Do the drugs used for chemotherapy are all very low response rate?

2) Authors should check the grammars in the article. For instance, Line 44, “Surgical resection of tumour and metastasis….”. The metastasis is not needed and it is confusing. It means “metastasis” is another treatment. Line 49, “cancellous” should be corrected to “cancerous”.

Methods:

1) Any reason why the concentration of EVs were used as the parameter for treatment instead of number of particles?

Results and Discussion:

1) Authors have indicated the total volume of 500 uL of EVs was isolated. However, what was the EVs protein concentration for fraction 7-9? If the concentration was diluted, more EVs in the form of PBS were added into the wells, resulting in the differential response if the volume of PBS was not standardised throughout the plate. Could authors provide more details on this? I concerned the effect was due to lack of media at high concentration in comparison to the control. For instance, 100 ug/mL FDNV, how many uL of PBS were actually added into the well? Was the same volume of PBS was added to the control? If the concentration of FDNV was diluted, end up the well has more PBS than the media. Fig 4A apoptosis, all the cells died, was it due to the PBS or FDNVs? The dose-dependent effect was then due to increasing volume of PBS or FDNVs? This has to be clarified.

2) Authors claimed that the EVs were successfully isolated from fingerroot. Although these were supported by size, however, the internal marker (E.g.: HSP70), external markers and the markers that are just present in plant but not in EVs should be shown. Although these have been well defined for mammals cells, authors should performed the best efforts to show the plant’s EVs markers.

3) Fig 2: Statistical analysis should be indicated in the figure.

4) Authors should discuss why there is a differential uptake in cancer cells in comparison to non-cancerous cells. What are the possible underlying mechanisms? This is interesting but no further detailed mechanisms were reported.

5) Although apoptosis was confirmed in the studied, however, the underlying mechanisms were not defined. Authors should at least investigated on certain pathway.

Reviewer #3: This work describes a method to isolate extracellular vesicles from Boesenbergia rotunda. The nanovesicles were further investigated for their anti-cancer properties on colon cancer cells. Based on in vitro results, the study concluded selective anti-cancer property of Boesenbergia rotunda derived nanovesicles.

Some comments as below:

1. The active constituent from Boesenbergia rotunda nanovesicles was not characterized / described

2. In lines 48-52, the authors specified that development of new anticancer agents for colorectal cancer is needed due to side effects in existing chemotherapy. However, the authors did not show the efficacy of Boesenbergia rotunda nanovesicles in comparison with existing chemotherapy.

3. In lines 67-69, the authors specified that limitation of crude Boesenbergia rotunda extract was non-specific cytotoxicity on non-cancerous cells. However, the authors did not compare the efficacy of crude Boesenbergia rotunda extract vs Boesenbergia rotunda nanovesicles in the context of selective cytotoxicity.

4. Thus, the purpose of developing Boesenbergia rotunda nanovesicles warrants further elaboration.

5. Unable to read Figure 3 due to low resolution.

6. The authors described Boesenbergia rotunda nanovesicles being similar to extracellular vesicles obtained from other edible plants. It will be informative to elaborate the similarities for the benefit of readers not familiar in this space.

7. ROS-induced apoptosis was demonstrated in colorectal cancer cells after treatment of Boesenbergia rotunda nanovesicles. It would be interesting to know, if similar mechanisms are observed in non-cancerous cells, i.e. the underlying mechanisms leading to the selective anti-cancer properties of Boesenbergia rotunda nanovesicles.

Reviewer #4: In this manuscript, authors isolated nanovesicles from fingerroot (FDNV) using differential centrifugation and size exclusion chromatography. Then, they used isolated FDNVs to treat two human colorectal cancer cell lines (HT-29 and HCT116) and one normal human colon cells line (CCD 841 CoN) and observed cytotoxicity in cancer cell lines but not in the normal cell line. Next, they investigated the uptake of FDNVs in all three cells lines. Following the confirmation of uptake of FDNVs in cancer cell lines, they examined the apoptosis percentage and the possible underlying mechanism that leads to apoptosis in both cancer cells line. Overall, I believe that the authors are off to a good start, however, several control experiments are missing (will explain in detail in the major comments section). This manuscript is within the scope of the journal and delivers a great scientific story. Hence, recommendation with major revision is advised.

Major comments:

1. In the introduction (and briefly in the discussion section, Line 413), authors introduced the concept of extracellular vesicles (EV). However, the whole manuscript is about FDNVs that are derived and isolated from the homogenate of fingerroot, which are not EVs. EVs are vesicles that are excreted (e.g., exosomes), hence, “extracellular” in the name. It is very misleading to have EV introduced in the introduction section and mentioned briefly in the discussion section, and it is wrong to equivalize FDNVs with EVs in the method section. Authors should make extinct differentiation in the manuscript between these two concepts.

2. For FDNV internalization, it seems the uptake was very localized for both cancer cell lines. Would that be the case for the normal cell line? It seems that there were way fewer number of cells in the frame for CCD 841 CoN comparing to the other two (based on DAPI staining) so that no uptake was captured under the microscope?

3. The apoptosis assay lacks the normal cell line control. If no apoptosis observed in normal cell line, then it will strengthen the conclusion of differential cytotoxicity and uptake in normal cell line.

4. Authors concluded that “FDNVs increased ROS generation and decreased GSH levels” in both CRC cell lines. Here, I think that this conclusion is a little premature and lacks two control experiments:

1) How did authors exclude the possibility that the increased intracellular ROS is not because/contaminated with the abundant ROS from peroxisome isolated from fingerroot that released into the cell through FDNV uptake? The differential centrifugation final pellets (after 100,000 *g) would contain all small membrane vesicles derived from fingerroot (lysosomes, endosomes, peroxisomes, microsomes, etc.) and all these vesicles are very similar in terms of size, so the SEC might not be sufficient to separate these organelles. Therefore, FDNVs are very likely to contain peroxisomes. A control experiment with just FDNVs (no cells) should be done to assess the extent of ROS contamination from FDNVs.

2) Both ROS and GSH measurement lacked a normal cell line control. Does the ROS and GSH amount stay the same in normal colon cell line? I understand that no cytotoxicity was observed in the normal colon cell line, but believes this control is necessary to strengthen the conclusions.

3) Additional question: why FDNV at 50 µg/mL generates more intracellular ROS but also have more cellular GSH than positive control hydrogen peroxide?

5. It is an interesting choice that the authors used µg/mL as their unit to describe the amount of FDNVs they treated the cells with. I assumed that this unit came from BCA assay, which is for total protein concentration. Is there a reason that the authors normalize all FDNVs treatment to total protein concentration? Is the potential active ingredient from fingerroot a protein? Does the total protein concentration of FDNV correlates with the # of FDNVs? The authors have NTA assay data, why not use the # FDNVs/mL?

6. All data should be reported in ± standard deviation (S.D.) instead of S.E.M. because you are reporting variabilities among your experiment replicates.

Minor comments:

1. Authors should state what medium (e.g., water, PBS, or etc.) they blended fingerroot in or no other liquid was added for homogenization in the method section.

2. Fig 1 C lacks statistical analysis.

3. Page 16, Line 353, please define PDNV in the discussion section (i.e., plant derived nano-vesicles (PDNV)).

4. Page 17 Line 367, differential centrifugation was used in this manuscript. Mentioning density gradient might confuse the reader. The authors should clarify to avoid confusion.

5. Page 17 Line 387-388, don’t need to capitalize pinostrobin, linoleic acid and phospholipase D.

6. Page 19 Line 435, add “chromatography” after “size exclusion”.

7. While the study appears to be sound, there are many typos, especially in introduction and discussion, making it difficult to follow. I advise the authors to re-read and revise the manuscript to improve the flow and readability of the text in introduction and discussion.

6. PLOS authors have the option to publish the peer review history of their article (what does this mean?). If published, this will include your full peer review and any attached files.

Reviewer #1: No

Reviewer #2: **Yes: **Jhi Biau Foo

Reviewer #3: No

Reviewer #4: No

---

## [Author Response · Author response to Decision Letter 0]

27 Feb 2022

Authors’ response to the editor and reviewers’ comments

Manuscript: PONE-D-21-31088

Title: Induction of apoptosis in human colorectal cancer cells by nanovesicles from fingerroot (Boesenbergia rotunda (L.) Mansf.)

Dear Editor,

Thank you for considering our manuscript for publication in PLOS ONE. We carefully read all the comments and suggestions from the editors and reviewers and revised the manuscript accordingly. Overall, we agree with the majority of the comments and revised the manuscript following the suggestions. 

What follows are our point-by-point responses to the comments from the editors and reviewers. Changes to the manuscript are also mentioned (track changes).

Journal Requirements:

Response: Thank you for your valuable suggestion. We have checked the templates and made the adjustments to meet the journal requirements.

Response: No permits were required for the described study, which complied with all relevant

regulations.

“This research project is supported by Mahidol University (Basic Research Fund: fiscal year 2021), Faculty of Science, Mahidol University, the Central Instrument Facility (CIF) Grant, Faculty of Science, Mahidol University and partially supported by Postdoctoral fellowship award from Mahidol University (grant number MD-PD_2021_12).”

Response: We have added "The funders had no role in study design, data collection and analysis, decision to publish, or preparation of the manuscript" in the cover letter.

Response: We have already uploaded study’s minimal underlying data set as Supporting Information files (S2 Table).

5. We note that you have included the phrase “data not shown” in your manuscript. Unfortunately, this does not meet our data sharing requirements. PLOS does not permit references to inaccessible data. We require that authors provide all relevant data within the paper, Supporting Information files, or in an acceptable, public repository. Please add a citation to support this phrase or upload the data that corresponds with these findings to a stable repository (such as Fig share or Dryad) and provide and URLs, DOIs, or accession numbers that may be used to access these data. Or, if the data are not a core part of the research being presented in your study, we ask that you remove the phrase that refers to these data.

Response: The data is not a core part of the research being presented in this study. We, therefore, removed the phrase “data not shown” that refers to these data from this revised manuscript. 

Response to Reviewers’ Comments:

Reviewer #1: 

1) I suggest the authors to analyze the fingerroot derived nanovesicle components by metabolomics, proteomics, and transcriptomics. 

Response: We appreciate for reviewer’s perspective. We have now performed the metabolomic analysis of FDNVs by LC-MS/MS. The method for LC-MS/MS has been incorporated in the Materials and Methods section (pages 6-7, lines 126-151). The distribution of the identified metabolites in FDNVs is shown in new Fig 1E. List of the discriminative putatively identified metabolites (58 named metabolites) of FDNVs is also presented in new S1 Table. Alkaloids were the most common FDNVs metabolite subtype (53%; 31/58), followed by phenolics (21%; 12/58), lipids (14%; 8/58), and organic compounds (12%; 7/58). Importantly, the phenolic compounds naringenin chalcone, pinostrobin, and pinocembrin were found in FDNVs. These phenolic compounds have been found as promising bioactive compounds in fingerroot juice [1]. Thus, these secondary metabolites may serve as medicinal agents that underlie the therapeutic action of FDNVs, which can improve our understanding of how FDNVs exhibit biological activities This detail information has been incorporated in the Result (page 13, lines 278-292). The discussion regarding the possible biological activity of the identified metabolites in FDNVs is now stated in the Discussion section (page 22, lines 498-506).

2) I suggest the authors to investigate the cellular internalization mechanism of fingerroot derived nanovesicles. A better understanding of fingerroot derived nanovesicle component and cellular uptake is essential to potentiate the capacity of nanovesicles to induce phenotypic changes in recipient cells.

Response: We thank the reviewer for the suggestion and we agree with the reviewer. In this revised manuscript, we investigated the uptake mechanism of FDNVs in colorectal cells. The uptake of FDNVs was examined in the presence of pinocytosis and phagocytosis inhibitors. As shown in new Fig 4, the uptake of FDNVs in HT29 and HCT116 were markedly inhibited by filipin (Fig 4A) and cytochalasin D (Fig 4B), which are the inhibitors of caveolae-mediated endocytosis and phagocytosis, respectively (P > 0.001). Conversely, treatment with amiloride, an inhibitor of micropinocytosis, and chlorpromazine, an inhibitor of clathrin-mediated endocytosis, did not affect the uptake of FDNVs in both cancer cell lines (Fig 4A). These data suggest that the internalization of FDNVs in colorectal cancer cells is partly via caveolae-mediated endocytosis and phagocytosis pathways. This related information has been incorporated in the Material and Methods (page 10, lines 218-228), Results (pages 16-17, Line 341-353 and 363-372) and Discussion (pages 23-24, lines 522-542) sections.

3) The introduction and discussion sections in this manuscript are also poorly constructed and written. Although their approach itself is of interest, the presentation does not suffice to show the novelty and significance that meet the standards of rigor required by the journal to be considered for publication.

Response: Thank you for your valuable comments. We have edited and revised the introduction and discussion of the manuscript. 

Reviewer #2

Introduction:

1) Line 47-48. The unsatisfactory response rate is referring to all the drugs used for colon cancer? Early or late stage or patient? This has to be clear otherwise it shows confusing statement. Do the drugs used for chemotherapy are all very low response rate?

Response: Thank you for your valuable suggestion. Chemotherapeutic drugs were applied to the high-risk stage II-IV CRC patients [2, 3]. Unfortunately, the overall response rate of advanced colorectal cancer to 5-fluorouracil (5-FU), the first-line drug, is only 10-15% [4]. Therefore, combining 5-FU, leucovorin, and capecitabine with oxaliplatin was recommended. However, a significant improvement in overall survival (OS) was observed only for stage III colon cancer [3, 5]. These data indicate an unsatisfactory response rate for colon cancer treatments. However, we revised the introduction in this manuscript according to reviewers #1 and #4 comments, emphasizing systemic toxicity to normal cells. Therefore, we design not to include the explanation of the unsatisfactory response rate mentioned in the previous version of the manuscript in this revised manuscript. 

2) Authors should check the grammars in the article. For instance, Line 44, “Surgical resection of tumour and metastasis….”. The metastasis is not needed and it is confusing. It means “metastasis” is another treatment. Line 49, “cancellous” should be corrected to “cancerous”.

Response: Thank you for your comments. The revised manuscript had been carefully checked to eliminate grammatical errors. In addition, we revised the introduction in this manuscript according to reviewers #1 and #4 comments; therefore, the sentence “Surgical resection of tumor and metastasis….” was removed. 

Methods:

1) Any reason why the concentration of EVs were used as the parameter for treatment instead of number of particles?

Response: We thank the reviewer for pointing this out. The particles/ml may represent the amount of EVs better than the total protein concentration (µg/ml) due to the possibility of other protein contamination in the sample. However, specific equipment like Nanoparticle tracking analysis (NTA) is required to measure the number of particles in the sample. Therefore, several laboratories, including us, measured the protein concentration using the BCA method to determine the concentration of EVs instead of the number of particles. Indeed, the total protein representing the concentration of EVs have been widely used to study the biological functions of plant-derived EVs; for example, EVs derived from citrus limon [6], ginger [7], and corn [8]. In addition, we found that the number of particles from isolated FDNVs correlated with the protein concentration (Fig 1B and S1 Fig); therefore, we used protein concentration as a parameter for treatment instead of particle number. 

Results and Discussion:

1) Authors have indicated the total volume of 500 uL of EVs was isolated. However, what was the EVs protein concentration for fraction 7-9? If the concentration was diluted, more EVs in the form of PBS were added into the wells, resulting in the plate. Could authors provide more details on this? I concerned the effect was due to lack of media at high concentration in comparison to the control. For instance, 100 ug/mL FDNV, how many uL differential response if the volume of PBS was not standardised throughout the of PBS were actually added into the well? Was the same volume of PBS was added to the control? If the concentration of FDNV was diluted, end up the well has more PBS than the media. Fig 4A apoptosis, all the cells died, was it due to the PBS or FDNVs? The dose-dependent effect was then due to increasing volume of PBS or FDNVs? This has to be clarified.

Response: Thank you for your valuable comments and suggestions. After ultracentrifugation, the vesicle pellet was resuspended with 1 ml PBS. Then, the FDNVs were purified using size exclusion chromatography (iZON). Fraction 8 (500 µl in PBS) was selected for all experiments based on the data of NTA and TEM, as mentioned in the result section (page12, lines 260-271). The protein concentration of this fraction was approximately 0.4 µg/ul (BCA protein assay), and total protein was 200 µg (500 µl x 0.4 µg/µl). 

To increase the concentration of FDNVs, we combined four vesicle pellets from ultracentrifugation before performing size exclusion chromatography. Thus, the total protein concentration was increased to approximately 1.5 µg/µl, and the total protein was 750 µg (500 µl x 1.5 µg/µl). These samples were used to study the activity of FDNVs. To investigate the effect of FDNVs on apoptosis induction, we prepared 100 µg/ml FDNVs by adding 67 µl of FDNVs (67 µl x 1.5 µg/µl = 100 µg) to 933 µl culture media and incubated with cells for 48 h. In the vehicle control group, we mixed 67 µl PBS with 933 µl culture media. We found no significant effect of PBS in the untreated condition (Fig 5). Therefore, the effect of 100 µg/ml FDNVs on apoptosis induction is not due to the dilution of the medium with PBS. This information has been incorporated in the Materials and Methods (page 8, line 178) section. 

2) Authors claimed that the EVs were successfully isolated from fingerroot. Although these were supported by size, however, the internal marker (E.g.: HSP70), external markers and the markers that are just present in plant but not in EVs should be shown. Although these have been well defined for mammal cells, authors should perform the best efforts to show the plant’s EVs markers.

Response: We thank the reviewer for pointing this out, and we agree with the reviewer. Although the guideline on the nomenclature and minimal information for studies of EVs have been recommended [9]. However, the markers guideline for plant-derived nanovesicles is currently unavailable due to insufficient information in the field of plant EVs [10]. In addition, the antibody specifically to plat proteins is also limited. However, as per your suggestion, we attempted to examine the expression of protein markers that were previously reported in mammalian cells-derived EVs, including TSG101, flotillin, CD81, by western blotting. In addition, calnexin was included as a negative marker for EVs. Unfortunately, we failed to detect the expression of all protein markers of mammalian cells-derived EVs in FDNVs (Figure 1 below). Therefore, these markers may not be specific for plant-derived EVs, or the antibodies failed to detect these plat proteins.

Figure 1: The expression of protein markers of mammalian cells-derived EVS in FDNVs by western blotting. 

3) Fig 2: Statistical analysis should be indicated in the figure.

Response: We thank the reviewer for the comment, we analyzed and provided the statistics in the revised Figure 2 and the result (pages 14-15, lines 297-322).

4) Authors should discuss why there is a differential uptake in cancer cells in comparison to non-cancerous cells. What are the possible underlying mechanisms? This is interesting but no further detailed mechanisms were reported.

Response: We agree with the reviewer and thank you for your suggestion. We found that CRC cells were more susceptible to FDNV uptake than normal colon cells (Fig 3). Moreover, the uptake of FDNVs in CRC cells was markedly inhibited by filipin (new Fig 4A) and cytochalasin D (new Fig 4B), which are the inhibitors of caveolae-mediated endocytosis and phagocytosis, respectively (P > 0.001). Conversely, treatment with amiloride, an inhibitor of micropinocytosis, and chlorpromazine, an inhibitor of clathrin-mediated endocytosis, did not affect the uptake of FDNVs in both CRC cell lines (new Fig 4A). These data indicate that the internalization of FDNVs in colorectal cancer cells is partly via caveolae-mediated endocytosis and phagocytosis pathways. However, these inhibitors did not affect the FDNVs internalization in normal colon epithelial cells (CCD 841 CoN) (Figure 2 below). This result indicates cell-dependent FDNVs uptake. 

Figure 2: The inhibition of FDNVs internalization in normal colon epithelial cells (CCD 841 CoN). ns: not significant. 

Indeed, the internalization of plant nanovesicles in cancer cells via phagocytosis and caveolae-mediated endocytosis mechanisms has been reported. For example, ginger-derived nano-lipids loaded with doxorubicin were mainly internalized via the phagocytosis pathway into CRC cancer cells that were significantly inhibited by cytochalasin D [7]. Moreover, the internalization of plant sap-derived extracellular vesicles breast cancer cells was mediated by phagocytosis and caveolae-mediated endocytosis [11]. Caveolin-1, the principal structural component of caveolae, is involved in caveolae-mediated endocytosis [12]. Although caveolin-1 function in cancer is controversial, overexpression of caveolin-1 has been reported in colon cancer [13]. Therefore, the caveolae-mediated endocytosis may be more effective in CRC, resulting in larger amounts of FDNVs internalization than in normal colon epithelial cells. In addition, the internalization of garlic-derived nanovesicles is mediated by interaction with the CD98 heavy chain (CD98hc) in liver cancer cells (HepG2) [14]. Expression levels of CD98hc protein were higher in CRC tissues than in matched normal tissues [15]. Therefore, upregulation of CD98hc might support the uptake of FDNVs in CRC cells. Taken together, these specific properties may help cancer to gain nanovesicles uptake inside the cells and explain the greater toxicity of FDNVs toward CRC cells. However, additional experiments are required to understand the FDNVs uptake mechanism in cancer cells. This information has been incorporated in the discussion (pages 23-24, lines 522-542).

5) Although apoptosis was confirmed in the studied, however, the underlying mechanisms were not defined. Authors should at least investigate on certain pathway.

Response: We completely agree with the reviewer. Intrinsic apoptosis pathway is characterized by mitochondria dysfunction-mediated cytochrome C release and subsequent activation of caspases-9 and caspases-3 [16]. In this revised manuscript, we determined the expression of the apoptosis-associated genes in HT-29 and HCT 116 cells treated with FDNVs by quantitative RT-PCR (new Fig 6). Treatment with FDNVs at 12.5 and 25 µg/ml markedly increased the expression of caspase-3 and caspase-9 in HT-29 and HCT116 cells (new Fig 6A and B). Aa increase in the expression of Bax, a pro-apoptotic gene, was also observed after treatment with 25 µg/ml FDNVs in both CRC cell lines. In contrast, the expression of Bcl-2, an anti-apoptotic gene, was decreased in both CRC cell lines after treatment with FDNVs at 25 µg/ml. These results indicate the anticancer effect of FDNVs was partially mediated through intrinsic apoptosis pathway. This information has been incorporated in the Materials and Methods (pages 8-9, lines 185-202), Results (pages 18-19, lines 403-412 and 421-426) and Discussion (page 24, lines 543-551) sections. 

Reviewer #3: 

1) The active constituent from Boesenbergia rotunda nanovesicles was not characterized / described

Response: We appreciate for reviewer’s perspective. We have now performed the metabolomic analysis of FDNVs by LC-MS/MS. The method for LC-MS/MS has been incorporated in the Materials and Methods (pages 6-7, lines 126-151) section. The distribution of the identified metabolites in FDNVs is shown in new Fig 1E. List of the discriminative putatively identified metabolites (58 named metabolites) of FDNVs is also presented in new S1 Table. Alkaloids were the most common FDNVs metabolite subtype (52%; 31/59), followed by phenolics (22%; 13/59), lipids (13%; 8/59), and organic compounds (12%; 7/59). Importantly, the phenolic compounds naringenin chalcone, pinostrobin, and pinocembrin were found in FDNVs. These phenolic compounds have been found as promising bioactive compounds in fingerroot juice [1]. Thus, these secondary metabolites may serve as medicinal agents that underlie the therapeutic action of FDNVs, which can improve our understanding of how FDNVs exhibit biological activities. This detail information has been incorporated in the Result (page 13, lines 278-284). The discussion regarding the possible biological activity of the identified metabolites in FDNVs is now stated in the Discussion (page 22, lines 498-506) section.

2) In lines 48-52, the authors specified that development of new anticancer agents for colorectal cancer is needed due to side effects in existing chemotherapy. However, the authors did not show the efficacy of Boesenbergia rotunda nanovesicles in comparison with existing chemotherapy.

Response: We appreciate the reviewer’s perspective. Currently, 5-fluorouracil (5-FU), a chemotherapeutic drug, was the first-line therapy for most CRC patients worldwide [17]. However, non-specific toxicity toward normal colon epithelial cells (CCD 841 CoN) of 5-FU has been reported [18]. Moreover, doxorubicin has been used as an adjuvant chemotherapy drug for CRC at advanced stages [19]. However, cytotoxicity of doxorubicin has also been reported against CCD 841 CoN [20]. This information indicates the non-selective cytotoxic effect of the conventional chemotherapeutic agents. Therefore, in this manuscript, we emphasize the selective activity of FDNVs toward CRC cells. Interestingly, we demonstrated that FDNVs exhibited cytotoxicity against CRC cells but not normal colon epithelial cells. Therefore, we did not compare the cytotoxicity of FDNVs with the conventional chemotherapeutic drugs. 

3) In lines 67-69, the authors specified that limitation of crude Boesenbergia rotunda extract was non-specific cytotoxicity on non-cancerous cells. However, the authors did not compare the efficacy of crude Boesenbergia rotunda extract vs Boesenbergia rotunda nanovesicles in the context of selective cytotoxicity. Thus, the purpose of developing Boesenbergia rotunda nanovesicles warrants further elaboration.

Response: We thank the reviewer for this suggestion. We compared the cytotoxic selectivity between fingerroot extract and its nanovesicles. In contrast to the selective cytotoxic effect of FDNVs, the fingerroot extract exhibited dose- and time-dependent effects against both cancer cells and normal human colon epithelial cells (Fig 2 and Table 2). Cytotoxicity of the fingerroot extract was significantly observed at 25 µg/ml after 24 h of treatment toward all tested cells (P < 0.001). Additionally, there was no difference between the IC50 values of the fingerroot extract against all tested cells. These results indicate the selective cytotoxic effect of FDNVs on colorectal cancer cells with relatively low cytotoxicity toward normal colon cells. This information has been incorporated in the Materials and Methods (page 8, line 167) and Results (pages 14-15, lines 297-322) sections.

4) Unable to read Figure 3 due to low resolution.

Response: With all due respect to the reviewer. We have improved the resolution of the figures. 

6) The authors described Boesenbergia rotunda nanovesicles being similar to extracellular vesicles obtained from other edible plants. It will be informative to elaborate the similarities for the benefit of readers not familiar in this space.

Response: We thank the reviewer for this valuable suggestion. We have added the result and discussion on the similarities of FDNVs to other nanovesicles from edible plants. This information has been incorporated in the Results (pages 12-13, lines 275-276) and Discussion (pages 21-23, lines 510-514, lines 528-530)

7) ROS-induced apoptosis was demonstrated in colorectal cancer cells after treatment of Boesenbergia rotunda nanovesicles. It would be interesting to know, if similar mechanisms are observed in non-cancerous cells, i.e. the underlying mechanisms leading to the selective anti-cancer properties of Boesenbergia rotunda nanovesicles.

Response: We thank the reviewer for this valuable suggestion. We have determined the effect of FDNVs on apoptosis induction in normal colon epithelial cells (CCD 841 CoN). As shown in new Fig 5C, there was no significant induction of early apoptosis in all tested concentrations of FDNVs compared to untreated control. Statistically significant induction of apoptosis was only found in the presence of 5% DMSO in human colon epithelial cells (32.2 ± 5.1 %, P < 0.001). More than 90% of cells remained viable even at a high concentration of FDNVs, indicating that FDNVs exhibited low cytotoxicity against normal colon cells. In contrast, treatment with 5% DMSO resulted in a significant reduction of cell viability (P < 0.001) relative to control. Late apoptosis and necrosis were not significantly different in all tested conditions. These results demonstrated that FDNVs displayed selective induction of apoptosis-mediated cell death in cancerous but not normal, cells. This information has been incorporated in the Results (pages 18-19, lines 391-402 and 414-419) sections.

Moreover, we determined the effect of FDNVs on the intracellular ROS level in normal colon epithelial cells (CCD 841 CoN). In contrast to CRC cells, only H2O2 causes significantly increased ROS (P < 0.05) and reduced GHS (P < 0.001) levels (new Fig. 7C). These data suggest that FDNVs showed selective cytotoxicity towards cancer cells through increased ROS production. This information has been incorporated in the Results (pages 19-21, lines 428-465) sections.

Reviewer #4

Major comments:

 1) In the introduction (and briefly in the discussion section, Line 413), authors introduced the concept of extracellular vesicles (EV). However, the whole manuscript is about FDNVs that are derived and isolated from the homogenate of fingerroot, which are not EVs. EVs are vesicles that are excreted (e.g., exosomes), hence, “extracellular” in the name. It is very misleading to have EV introduced in the introduction section and mentioned briefly in the discussion section, and it is wrong to equivalize FDNVs with EVs in the method section. Authors should make extinct differentiation in the manuscript between these two concepts.

Response: Thank you for your valuable suggestion. We agree that EVs and nanovesicles (NVs) have different concepts. According to the recent report [10], the term “plant-derived nanovesicles (PDNV)” is suggested for vesicular fractions obtained from plant tissues when destructive processes are used and when natural release into the extracellular space cannot be established. Therefore, based on the method used to isolate vesicles, “EVs” is now changed to “nanovesicles (NVs)” throughout the manuscript.

2) For FDNV internalization, it seems the uptake was very localized for both cancer cell lines. Would that be the case for the normal cell line? It seems that there were way fewer number of cells in the frame for CCD 841 CoN comparing to the other two (based on DAPI staining) so that no uptake was captured under the microscope?

Response: We thank the reviewer for pointing this out, and we agree with the reviewer. The size of CCD 841 CoN cells is bigger than CRC cells; therefore, a low number of cells was observed under the same magnification. However, we quantified the fluorescence intensity using the same number of cells as shown in Fig 3D. 

3) The apoptosis assay lacks the normal cell line control. If no apoptosis observed in normal cell line, then it will strengthen the conclusion of differential cytotoxicity and uptake in normal cell line.

Response: We thank the reviewer for this valuable suggestion. We have determined the effect of FDNVs on apoptosis induction in normal colon epithelial cells (CCD 841 CoN). As shown in new Fig 5C, there was no significant induction of early apoptosis in all tested concentrations of FDNVs compared to untreated control. Statistically significant induction of apoptosis was only found in the presence of 5% DMSO in human colon epithelial cells (32.2 ± 5.1 %, P < 0.001). More than 90% of cells remained viable even at a high concentration of FDNVs, indicating that FDNVs exhibited low cytotoxicity against normal colon cells. In contrast, treatment with 5% DMSO resulted in a significant reduction of cell viability (P < 0.001) relative to control. Late apoptosis and necrosis were not significantly different in all tested conditions. These results demonstrated that FDNVs displayed selective induction of apoptosis-mediated cell death in cancerous but not normal, cells. This information has been incorporated in the Results (pages 18-19, lines 391-402 and 414-419) sections.

4) Authors concluded that “FDNVs increased ROS generation and decreased GSH levels” in both CRC cell lines. Here, I think that this conclusion is a little premature and lacks two control experiments:

4.1) How did authors exclude the possibility that the increased intracellular ROS is not because/contaminated with the abundant ROS from peroxisome isolated from fingerroot that released into the cell through FDNV uptake? The differential centrifugation final pellets (after 100,000 *g) would contain all small membrane vesicles derived from fingerroot (lysosomes, endosomes, peroxisomes, microsomes, etc.) and all these vesicles are very similar in terms of size, so the SEC might not be sufficient to separate these organelles. Therefore, FDNVs are very likely to contain peroxisomes. A control experiment with just FDNVs (no cells) should be done to assess the extent of ROS contamination from FDNVs.

Response: We thank the reviewer for this constructive suggestion. We have additionally determined the ROS level in FDNVs (Figure 3 below). Briefly, 12.5-50 mg/ml FDNVs were stained with CM-H2DCFDA for 30 min. HT-29 cells treated with 50 mg/ml FDNVs and 200 mM H2O2 were positive controls. Then the fluorescence signal was determined using EnVision® multimode plate reader (Ex/Em: ∼492–495/517–527 nm). We found that treatments with 50 mg/ml FDNVs (P < 0.001) and 200 mM H2O2 (P < 0.001) significantly induced intracellular ROS levels in HT29 when compared with untreated cells. However, no fluorescent signal was detected in FDNVs. These results indicate no ROS contamination from FDNVs. 

Figure 3: Induction of intracellular ROS level in FDNVs-treated normal colon epithelial cells (CCD 841 CoN). **P < 0.01, ***P < 0.001 (one-way ANOVA).

4.2) Both ROS and GSH measurement lacked a normal cell line control. Does the ROS and GSH amount stay the same in normal colon cell line? I understand that no cytotoxicity was observed in the normal colon cell line, but believes this control is necessary to strengthen the conclusions.

Response: We thank the reviewer for pointing this out, and we agree with the reviewer. We determined the effect of FDNVs on the intracellular ROS and GSH levels in normal colon epithelial cells (CCD 841 CoN). In contrast to CRC cells, only H2O2 causes significantly increased ROS (P < 0.05) and reduced GHS (P < 0.001) levels (new Fig 7C). These data further support the selective cytotoxicity of FDNVs towards cancer cells through increased ROS production. This information has been incorporated in the Results (pages 19-21, lines 428-465) sections.

4.3) Additional question: why FDNV at 50 µg/mL generates more intracellular ROS but also have more cellular GSH than positive control hydrogen peroxide?

Response: We thank the reviewer for pointing this out. The cellular antioxidant mechanisms play critical roles in protecting the cells and organisms from oxidative damage. Several antioxidant mechanisms, both enzymatic and non-enzymatic systems, have been reported to counteract oxidative stress in human cells and organisms [21]. Moreover, several antioxidant mechanisms of Boesenbergia rotunda have been reported [22]. However, other antioxidant mechanisms may involve oxidative stress induced by FDNVs, which is different from the oxidative stress induced by H2O2. Therefore, further experiments may be required to study the effects of FDNVs on oxidative stress defense mechanisms. 

5) It is an interesting choice that the authors used µg/mL as their unit to describe the amount of FDNVs they treated the cells with. I assumed that this unit came from BCA assay, which is for total protein concentration. Is there a reason that the authors normalize all FDNVs treatment to total protein concentration? Is the potential active ingredient from fingerroot a protein? Does the total protein concentration of FDNV correlates with the # of FDNVs? The authors have NTA assay data, why not use the # FDNVs/mL?

Response: We thank the reviewer for pointing this out. The particles/ml may represent the amount of EVs better than the total protein concentration (µg/ml) due to the possibility of other protein contamination in the sample. However, specific equipment like Nanoparticle tracking analysis (NTA) is required to measure the number of particles in the sample. Therefore, several laboratories, including us, measured the protein concentration using the BCA method to determine the concentration of EVs instead of the number of particles. Indeed, the total protein representing the concentration of EVs have been widely used to study the biological functions of plant-derived EVs; for example, EVs derived from citrus limon [6], ginger [7], and corn [8]. In addition, we found that the number of particles from isolated FDNVs correlated with the protein concentration (Fig 1B and S1 Fig); therefore, we used protein concentration as a parameter for treatment instead of particle number. 

6) All data should be reported in ± standard deviation (S.D.) instead of S.E.M. because you are reporting variabilities among your experiment replicates.

Response: Thank you for your suggestion. All data are now reported as means ± standard deviation (SD).

Minor comments:

1. Authors should state what medium (e.g., water, PBS, or etc.) they blended fingerroot in or no other liquid was added for homogenization in the method section.

Response: There was no liquid during the blending process. We have incorporated this information in the section of Materials and Methods (page 4, lines 86-88). 

2. Fig 1 C lacks statistical analysis.

Response: Fig 1C (zeta potential) was changed to Fig 1D in the revised manuscript. In addition, we have performed the statistical analysis of Fig 1D.

3. Page 16, Line 353, please define PDNV in the discussion section (i.e., plant derived nano-vesicles (PDNV)).

Response: We have revised according to your suggestion (page 21, lines 479). 

4. Page 17 Line 367, differential centrifugation was used in this manuscript. Mentioning density gradient might confuse the reader. The authors should clarify to avoid confusion.

Response: We thank the reviewer for your suggestion. In this revised manuscript, “density gradient centrifugation-based methods” is now replaced with “sucrose density-gradient separation methods”. (page 22, lines 494-495). 

5. Page 17 Line 387-388, don’t need to capitalize pinostrobin, linoleic acid and phospholipase D.

Response: We have corrected it according to your suggestion (page 24, lines 557). 

6. Page 19 Line 435, add “chromatography” after “size exclusion”.

Response: We have deleted this phrase according to revision. 

7. While the study appears to be sound, there are many typos, especially in introduction and discussion, making it difficult to follow. I advise the authors to re-read and revise the manuscript to improve the flow and readability of the text in introduction and discussion.

Response: We thank the reviewer for this constructive comment. We have edited and revised the introduction and discussion of the manuscript.

End of Response to the Reviewers

We appreciate the reviewers for not only their time but also their constructive and helpful comments, which helped us improve our manuscript. In addition to the changes described above, we noticed and fixed minor errors from our originally submitted version. We also carefully re-read the manuscript and made some further minor changes to improve clarity and readability. We think the manuscript is now ready for publication, yet please do not hesitate to contact us should any further questions arise or if you feel additional corrections are necessary.

Sincerely,

Nittaya Boonmuen on behalf of all authors

References

1. Chahyadi A, Hartati R, Wirasutisna KR. 2014. Boesenbergia pandurata Roxb., an Indonesian medicinal plant: Phytochemistry, biological activity, plant biotechnology. Procedia Chem 13:13-37. https://doi.org/10.1016/j.proche.2014.12.003

2. Labianca R, Nordlinger B, Beretta G, Mosconi S, Mandalà M, Cervantes A, et al. 2013. Early colon cancer: ESMO Clinical Practice Guidelines for diagnosis, treatment and follow-up. Annals of oncology 24:vi64-vi72. https://10.1093/annonc/mdt354 PMID: 24078664

3. Argilés G, Tabernero J, Labianca R, Hochhauser D, Salazar R, Iveson T, et al. 2020. Localised colon cancer: ESMO Clinical Practice Guidelines for diagnosis, treatment and follow-up. Annals of Oncology 31:1291-305. https://10.1016/j.annonc.2020.06.022 PMID: 32702383

4. Giacchetti S, Perpoint B, Zidani R, Le Bail N, Faggiuolo R, Focan C, et al. 2000. Phase III multicenter randomized trial of oxaliplatin added to chronomodulated fluorouracil–leucovorin as first-line treatment of metastatic colorectal cancer. Journal of clinical oncology 18:136-. https://10.1200/JCO.2000.18.1.136 PMID: 10623704

5. Douillard J, Cunningham D, Roth A, Navarro M, James R, Karasek P, et al. 2000. Irinotecan combined with fluorouracil compared with fluorouracil alone as first-line treatment for metastatic colorectal cancer: a multicentre randomised trial. Lancet 355:1041-7. https://10.1016/s0140-6736(00)02034-1 PMID: 10744089

6. Raimondo S, Naselli F, Fontana S, Monteleone F, Dico AL, Saieva L, et al. 2015. Citrus limon-derived nanovesicles inhibit cancer cell proliferation and suppress CML xenograft growth by inducing TRAIL-mediated cell death. Oncotarget 6:19514-27. https://doi.org/10.18632/oncotarget.4004 PMID: 26098775

7. Zhang M, Xiao B, Wang H, Han MK, Zhang Z, Viennois E, et al. 2016. Edible ginger-derived nano-lipids loaded with doxorubicin as a novel drug-delivery approach for colon cancer therapy. Mol Ther 24:1783-96. https://10.1038/mt.2016.159 PMID: 27491931

8. Sasaki D, Kusamori K, Takayama Y, Itakura S, Todo H, Nishikawa M. 2021. Development of nanoparticles derived from corn as mass producible bionanoparticles with anticancer activity. Sci Rep 11:1-12. https://10.1038/s41598-021-02241-y PMID: 34819568

9. Théry C, Witwer KW, Aikawa E, Alcaraz MJ, Anderson JD, Andriantsitohaina R, et al. 2018. Minimal information for studies of extracellular vesicles 2018 (MISEV2018): a position statement of the International Society for Extracellular Vesicles and update of the MISEV2014 guidelines. J Extracell Vesicles 7:1535750. https://10.1080/20013078.2018.1535750 PMID: 30637094

10. Pinedo M, de la Canal L, de Marcos Lousa C. 2021. A call for Rigor and standardization in plant extracellular vesicle research. J Extracell Vesicles 10:e12048. https://doi.org/10.1002/jev2.12048 PMID: 33936567

11. Kim K, Yoo HJ, Jung J-H, Lee R, Hyun J-K, Park J-H, et al. 2020. Cytotoxic effects of plant sap-derived extracellular vesicles on various tumor cell types. J Funct Biomater 11:22. https://doi.org/10.3390/jfb11020022 PMID: 32252412

12. Yang C, He B, Dai W, Zhang H, Zheng Y, Wang X, et al. 2021. The role of caveolin-1 in the biofate and efficacy of anti-tumor drugs and their nano-drug delivery systems. Acta Pharm Sin B 11:961-77. https://10.1016/j.apsb.2020.11.020 PMID: 33996409

13. Patlolla JM, Swamy MV, Raju J, Rao CV. 2004. Overexpression of caveolin-1 in experimental colon adenocarcinomas and human colon cancer cell lines. Oncol Rep 11:957-63. https://doi.org/10.3892/or.11.5.957 PMID: 15069532

14. Song H, Canup BS, Ngo VL, Denning TL, Garg P, Laroui H. 2020. Internalization of garlic-derived nanovesicles on liver cells is triggered by interaction with CD98. ACS omega 5:23118-28. https://10.1021/acsomega.0c02893 PMID: 32954162

15. Lee D, Kim HS, Kim HU, Song HJ, Lee C, Chun HM, et al. 2022. Expression profile of CD98 heavy chain and L-type amino acid transporter 1 and its prognostic significance in colorectal cancer. Pathol Res 229:153730. https://10.1016/j.prp.2021.153730 PMID: 34942513

16. Elmore S. 2007. Apoptosis: a review of programmed cell death. Toxicol Pathol 35:495-516. https://10.1080/01926230701320337 PMID: 17562483

17. Cunningham D, Sirohi B, Pluzanska A, Utracka-Hutka B, Zaluski J, Glynne-Jones R, et al. 2009. Two different first-line 5-fluorouracil regimens with or without oxaliplatin in patients with metastatic colorectal cancer. Annals of oncology 20:244-50. https://10.1093/annonc/mdn638 PMID: 18854549

18. Koosha S, Mohamed Z, Sinniah A, Alshawsh MA. 2019. Investigation into the Molecular Mechanisms underlying the Anti-proliferative and Anti-tumorigenesis activities of Diosmetin against HCT-116 Human Colorectal Cancer. Sci Rep 9:1-17. https://10.1038/s41598-019-41685-1 PMID: 30914796

19. Weinländer G, Kornek G, Raderer M, Hejna M, Tetzner C, Scheithauer W. 1997. Treatment of advanced colorectal cancer with doxorubicin combined with two potential multidrug-resistance-reversing agents: high-dose oral tamoxifen and dexverapamil. Journal of cancer research and clinical oncology 123:452-5. https://10.1007/BF01372550 PMID: 9292709

20. Mendez-Encinas MA, Carvajal-Millan E, Rascón-Chu A, Astiazarán-García H, Valencia-Rivera DE, Brown-Bojorquez F, et al. 2019. Arabinoxylan-based particles: In vitro antioxidant capacity and cytotoxicity on a human colon cell line. Medicina 55:349. https://10.3390/medicina55070349 PMID: 31284672

21. Kennedy L, Sandhu JK, Harper M-E, Cuperlovic-Culf M. 2020. Role of glutathione in cancer: From mechanisms to therapies. Biomolecules 10:1429. https://10.3390/biom10101429 PMID: 33050144

22. Eng-Chong T, Yean-Kee L, Chin-Fei C, Choon-Han H, Sher-Ming W, Li-Ping CT, et al. 2012. Boesenbergia rotunda: from ethnomedicine to drug discovery. Evid Based Complement Alternat Med 2012:1-25. https://doi.org/10.1155/2012/473637 PMID: 23243448

---

## [Decision Letter · Decision Letter 1]

14 Mar 2022

Induction of apoptosis in human colorectal cancer cells by nanovesicles from fingerroot (Boesenbergia rotunda (L.) Mansf.)

PONE-D-21-31088R1

Dear Dr. Boonmuen,

We’re pleased to inform you that your manuscript has been judged scientifically suitable for publication and will be formally accepted for publication once it meets all outstanding technical requirements.

Kind regards,

Lay-Hong Chuah

Academic Editor

PLOS ONE

Additional Editor Comments (optional):

Reviewers' comments:

Reviewer's Responses to Questions

**Comments to the Author**

1. If the authors have adequately addressed your comments raised in a previous round of review and you feel that this manuscript is now acceptable for publication, you may indicate that here to bypass the “Comments to the Author” section, enter your conflict of interest statement in the “Confidential to Editor” section, and submit your "Accept" recommendation.

Reviewer #2: All comments have been addressed

Reviewer #3: All comments have been addressed

Reviewer #4: All comments have been addressed

2. Is the manuscript technically sound, and do the data support the conclusions?

Reviewer #2: Yes

Reviewer #3: Yes

Reviewer #4: (No Response)

3. Has the statistical analysis been performed appropriately and rigorously? 

Reviewer #2: Yes

Reviewer #3: Yes

Reviewer #4: (No Response)

4. Have the authors made all data underlying the findings in their manuscript fully available?

Reviewer #2: Yes

Reviewer #3: (No Response)

Reviewer #4: (No Response)

5. Is the manuscript presented in an intelligible fashion and written in standard English?

Reviewer #2: Yes

Reviewer #3: Yes

Reviewer #4: (No Response)

6. Review Comments to the Author

Reviewer #2: (No Response)

Reviewer #3: (No Response)

Reviewer #4: (No Response)

7. PLOS authors have the option to publish the peer review history of their article (what does this mean?). If published, this will include your full peer review and any attached files.

Reviewer #2: No

Reviewer #3: No

Reviewer #4: No

---

## [Editor Report · Acceptance letter]

24 Mar 2022

PONE-D-21-31088R1 

Induction of apoptosis in human colorectal cancer cells by nanovesicles from fingerroot (Boesenbergia rotunda (L.) Mansf.) 

Dear Dr. Boonmuen:

I'm pleased to inform you that your manuscript has been deemed suitable for publication in PLOS ONE. Congratulations! Your manuscript is now with our production department. 

Kind regards, 

on behalf of

Dr. Lay-Hong Chuah 

Academic Editor

PLOS ONE